# A deep state-space analysis framework for cancer patient latent state estimation and classification from EHR time-series data

Yuji Okamoto[1], Aya Nakamura[1], Ryosuke Kojima [1], Eiichiro Uchino[1,2], Yohei Mineharu[1,3], Yohei Harada[1], Mayumi Kamada[1], Minoru Sakuragi[1,2], Manabu Muto[4], Motoko Yanagita[2,5], Yasushi Okuno[1]*

**1** Department of Biomedical Data Intelligence, Graduate School of Medicine, Kyoto University, Kyoto, Japan, **2** Department of Nephrology, Graduate School of Medicine, Kyoto University, Kyoto, Japan, **3** Department of Artificial Intelligence in Healthcare and Medicine, Graduate School of Medicine, Kyoto University, Kyoto, Japan, **4** Department of Medical Oncology, Graduate School of Medicine, Kyoto University, Kyoto, Japan, **5** Institute for the Advanced Study of Human Biology (ASHBi), Kyoto University, Kyoto, Japan

☯ These authors contributed equally to this work.
* okuno.yasushi.4c@kyoto-u.ac.jp

## Abstract

Advancements in deep learning technologies and an increase in medical data have enhanced the accuracy of disease diagnosis and treatment strategies. Notably, significant progress has been made in the use of deep learning-based time-series prediction models for short-term disease onset prediction and analysis of important features. However, research on explainable deep learning for long-term disease progression, such as cancer and chronic diseases, still faces challenges. The difficulty in estimating explainable gradual disease progression from observable patient test data is a key factor. To address this issue, we propose a new approach called the "deep state-space analysis framework." This framework utilizes sequentially obtained electronic health records (EHRs) to estimate and visualize temporal changes in the latent states of patients related to disease progression. It enables the clustering of latent patient states according to the severity of disease progression and identifies key factors leading to a poor prognosis with medication. To validate our framework, a detailed analysis of data from 12,695 patients with cancer was conducted. The estimated transitions of the latent states capture the clinical status of the patients and their continuous temporal changes. Furthermore, anemia was identified as a poor prognostic factor during state transitions in patients with cancer. Significant features were also confirmed, such as immune cell abnormalities, which are poor prognostic factors in patients treated with Nivolumab, Osimertinib, and Afatinib. This technological innovation deepens our understanding of disease progression and supports early treatment adjustments, prognostic evaluations, and the formulation of optimal

**Data availability statement:** Data cannot be shared publicly because of patient privacy in electronic medical records. Data are available from the Kyoto University Graduate School and Faculty of Medicine, Ethics Committee (contact via email: ethcom@kuhp.kyoto-u.ac.jp, telephone: +81-75-753-4680, or website: http://www.ec.med.kyoto-u.ac.jp/) for researchers who meet the criteria for access to confidential data.

**Funding:** This work was supported by JST, Center of Innovation Program (JPMJCE1302), awarded to RK, EU, YM, MS, YO. This work was supported by JST Moonshot R&D Grant Number (JPMJMS2021), awarded to YO. This work was supported by JST Moonshot R&D Grant Number (JPMJMS2024), awarded to RK. The funders had no role in study design, data collection and analysis, decision to publish, or preparation of the manuscript.

**Competing interests:** The authors have declared that no competing interests exist.

long-term strategies. With the advancements in deep learning, its application in healthcare has even greater potential.

## Introduction

The application of deep learning to electronic health records (EHR) [1–3] is facilitating the construction of new medical systems such as personalized medicine and real-time medical care. Expressive deep learning is well-modeled for complex EHR data, demonstrating high predictive accuracy. In the healthcare sector, the construction of interpretable deep learning models and estimation of disease causes are advancing [4–6]. Interpretability studies have been applied to instantaneous test data and medical images, and research to estimate disease factors caused by sequential changes in disease progression is still developing.

In predicting disease progression using sequential EHR data, a deep learning model that uniformly handles heterogeneous data structures is being developed [7,8]. In particular, Long Short-Term Memory (LSTM) [9] and Transformers [10], which are deep time-series models, have been used to account for the effects of sequential data. These deep time-series models can naturally learn long-term changes in diseases, enabling medical event prediction and the classification of disease subtypes. Therefore, using deep time-series models, we could translate long-term disease behavior into the dynamics of interpretable latent states of patients and construct a time-series-based treatment strategy, making the underlying disease progression more interpretable and accessible.

The estimation of latent states assumes the existence of latent states that cannot be directly observed, such as the medical condition of the patient, but that can influence the observed data. This method involves estimating latent states from the behavior of the observed data. In conventional methods of latent state estimation, the direct representation of temporal changes within the latent state-space has not been considered. Instead, deep time-series models that directly learn the observed time-series data are widely used. Moreover, owing to the black-box nature of deep learning, many evaluations have focused more on observed states, such as prognosis, rather than latent states, and the consideration of the clinical interpretability of the estimated latent states and the temporal risk factors represented therein has been limited [11,12].

To overcome the limitations, we have developed a new framework called the "Deep state-space analysis framework" to enable the clinical interpretation of patients and the identification of temporal risk factors in the latent state-space by explicitly modeling the temporal changes of patient latent states (Fig 1). In this framework, we utilize time-series observational data recorded in EHR to perform unsupervised learning of the temporal changes in patients' observed states using a "deep state-space model" [13]. This model employs deep neural networks (DNN) to represent temporal changes in latent states and the relationship between the observed and latent states in a time-series model. In particular, a deep state-space model can capture temporal

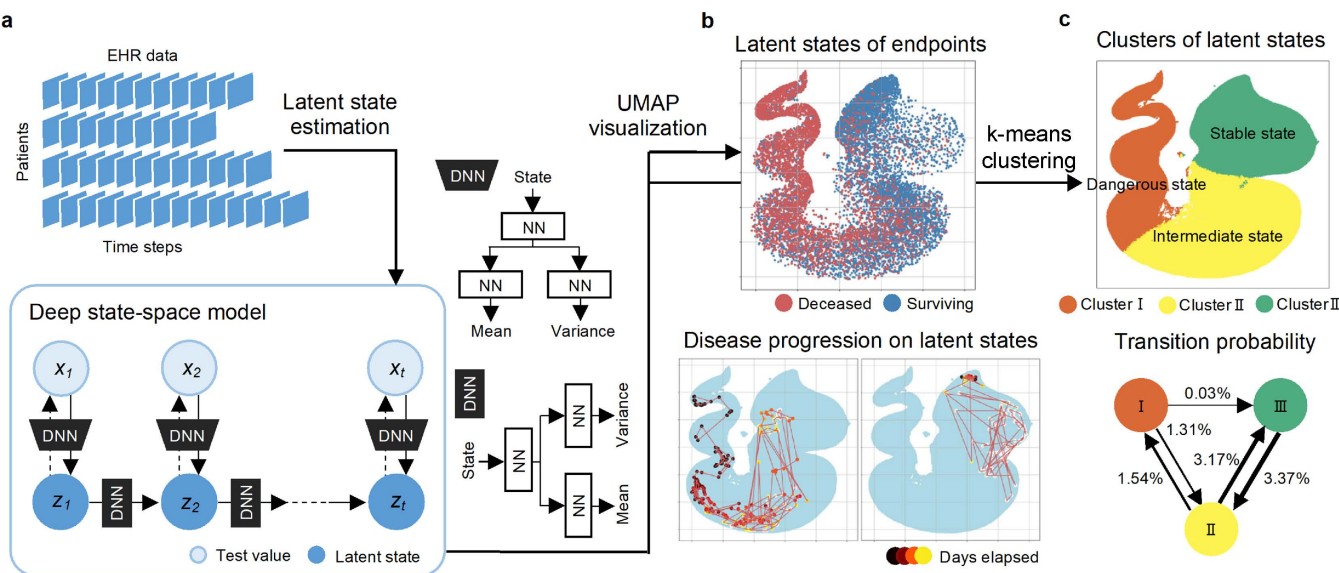

**Fig 1. Deep state-space analysis framework overview. (a)** An overview of the deep state-space model used for latent state estimation in the context of time-series EHR is shown. The deep state-space model is a DNN-based time-series prediction model, and it provides an example configuration for each DNN. Here, $x_t$ represents the observed state at time t (t = 1, 2, …, T), indicating the input of various test items from the time-series EHR. $z_t$ represents the latent state at time t (t = 1, 2, …, T), and the output yields a latent state represented in multiple dimensions. **(b)** Visualization of latent states is conducted using UMAP. The upper part of (b) shows the distribution of surviving and deceased patients in the latent space, while the lower part shows the typical trajectories for each group. **(c)** Stratification of patient states is performed using k-means. Each cluster is classified into one of three categories: Dangerous State, Intermediate State, and Stable State, and their transition probabilities are shown at the bottom of (c).

transitions using a DNN, allowing it to learn temporal transitions from data and capture spatiotemporal continuity. In our framework, the visualization of the latent state-space learned by the deep state-space model is performed using Uniform Manifold Approximation and Projection (UMAP) [14] for each time point of the patient, associating it with disease progression, thus enabling the clinical interpretation of latent states. Furthermore, by applying k-means clustering to the latent state of each patient, statistically significant observed variables (medical tests) specific to each cluster were identified, allowing the identification of the characteristic observed variables (medical tests) for each cluster.

In this study, we applied our deep state-space analysis framework to the EHR of 12,695 patients with cancer who underwent chemotherapy at Kyoto University Hospital. A time-series latent state estimation was conducted for patients with cancer, successfully evaluated the clinical interpretability of the latent states, and identified temporal risk factors. To the best of our knowledge, this study is the first to succeed in capturing the clinical validity and temporal risk factors in time-series latent state estimation by modeling changes over time. This framework, using a scalable deep learning-based deep state-space model, allows for the construction of models without limiting specific types of diseases or treatments. By directly modeling time-series changes, it enables the analysis of patient states over the long term and captures disease progression. In the future, leveraging this framework for chronic diseases that require long-term monitoring, not limited to cancer, holds the potential to advance patient health management using latent states, optimize treatment strategies, such as early treatment modification and prognosis assessment, and contribute to further advancements in healthcare.

Our main contributions are following:

- We develop a deep state-space analysis framework for estimating and visualizing clinically interpretable latent states from EHR time-series data of chemotherapy patients.

- We identify three distinct latent states (dangerous, intermediate, and stable) and their temporal relationships, revealing important temporal risk factors in cancer patients treated with anticancer drugs.

- We demonstrate drug-specific factors by identifying characteristic test items during state transitions for specific anticancer drugs.

## Methods

In this study, we used observable patient data from EHR to estimate latent states using a deep state-space model that considers the temporal sequence. We interpreted the obtained latent states and evaluated (a) the clinical validity of the latent states and (b) the prognostic risk for patients.

First, for (a) the clinical validity of the latent states, we investigated the correlation between prognosis and latent states using the latent states of the deceased and surviving patients. We assumed that the time-series of the deceased patients would have a final state closer to death within the temporal EHR sequence, representing an adverse prognostic state. In contrast, we considered that the final state in the time-series of the surviving patients would be relatively favorable. Based on this, we examined the distribution of the final state in the time-series of latent states for the deceased and surviving patients to confirm whether the latent states captured the states associated with the prognosis. In addition, we compared this confirmation of latent states with other latent state estimation methods to verify the effectiveness of our approach concerning (a).

Next, we clustered the latent states obtained through state estimation and evaluated (b) the prognostic risk for patients by identifying clusters with a high proportion of deceased patients. Specifically, to discover the temporal risk factors related to the prognosis of patients receiving eight anticancer drugs, we estimated the transition probabilities between the clusters of deceased and surviving patients. We quantitatively assessed the temporal transitions of the patient states and interpreted the clusters. Furthermore, by investigating test items with significant differences in distribution between clusters, we explored the temporal risk factors that are crucial during latent state transitions.

### Ethics review

This study was approved by the Ethics Committee of Kyoto University Hospital (Approval No.: R1498). Given that the data employed in this study comprised only existing information obtained from clinical practice, the study was conducted using the opt-out method under Japanese law. We posted an announcement regarding this study on our hospital and department websites and provided information on exclusion from participation.

Due to the retrospective nature of the study, Kyoto University Graduate School and Faculty of Medicine, Ethics Committee waived the need of obtaining informed consent in accordance with Japanese laws and regulations.

### Dataset

The dataset used in this study was constructed using electronic medical record (EMR) data from Kyoto University Hospital, a tertiary teaching hospital in Japan, covering the period from January 1, 2006, to October 31, 2018. These data were obtained on May 20, 2020. The dataset was generated and reviewed based on clinical information derived from the institution's EMRs, and this study was conducted in accordance with the principles of the Declaration of Helsinki. Only data obtained during routine medical practice were used, and informed consent was obtained on an opt-out basis in accordance with Japanese laws and regulations. All explanations of the study and expressions of consent were provided in written form, ensuring that participants received comprehensive information and that their agreement or disagreement was properly documented. This approach was approved by the Ethics Committee of Kyoto University as a valid method of consent for a study of this nature. To ensure ethical compliance, detailed information regarding the purpose of the study, the nature of the data used, and the right of participants to withdraw was made publicly available on the

Kyoto University Hospital website (https://www.kuhp.kyoto-u.ac.jp/outline/research-disclosure.html), thereby safeguarding participants' autonomy. This study received approval from the Kyoto University Ethics Committee (Approval Number R1498), recognizing its suitability as a retrospective investigation. All data analysis for this study was completed by March 31, 2024. Throughout the data collection process, the authors did not access any information that could identify individual participants.

Eligible patients were patients with cancer aged 20 years or older who underwent serum creatinine measurement as part of a routine blood test and received at least one oral or injectable anticancer drug. Patients with no data or a short observation series length (time steps) were excluded, specifically those with fewer than 50 steps, based on the distribution of the number of steps (S1 Appendix). In total, 12,695 patients were included in the analysis. A comprehensive summary of the dataset is presented in Table 1; if a patient had more than one cancer type, there may be an overlap in cancer types. For the classification of cancer types, the ICD-10 intermediate item groups for neoplasms were used [15].

**Table 1. Demographic and clinical characteristics of the patient population. The table presents the number of patients in the overall study cohort (All; N = 12,695) and the deceased subgroup (Dead; N = 4,668). Patients are stratified by age group, gender, and primary diagnosis according to the International Classification of Diseases, 10th Revision (ICD-10).**

| | | All (N = 1,2695) | Dead (N = 4,668) |
|---|---|---|---|
| Demographics | | | |
| Age-group | 20-39 years old | 561 | 185 |
| | 40-59 years old | 2,673 | 1,003 |
| | 60-79 years old | 7,792 | 3,078 |
| | 80- years old | 1,669 | 402 |
| Gender | Male | 7,180 | 2,867 |
| | Female | 5,515 | 1,801 |
| Disease cohort | | | |
| Cancer type (ICD-10 classification) | C00-14 (Malignant neoplasms of lip, oral cavity and pharynx) | 1151 | 348 |
| | C15-26 (Malignant neoplasms of digestive organs) | 6978 | 2700 |
| | C30-39 (Malignant neoplasms of respiratory and intra-thoracic organs) | 4874 | 2032 |
| | C45-49 (Malignant neoplasms of mesothelial and soft tissue) | 963 | 384 |
| | C50 (Malignant neoplasm of breast) | 1540 | 318 |
| | C51-58 (Malignant neoplasms of female genital organs) | 2171 | 614 |
| | C60-63 (Malignant neoplasms of male genital organs) | 2331 | 605 |
| | C64-68 (Malignant neoplasms of urinary tract) | 2105 | 621 |
| | C69-72 (Malignant neoplasms of eye, brain and other parts of central nervous system) | 342 | 151 |
| | C73-75 (Malignant neoplasms of thyroid and other endocrine glands) | 503 | 128 |
| | C76-80 (Malignant neoplasms of ill-defined, secondary and unspecified sites) | 8554 | 3448 |
| | C81-96 (Malignant neoplasms, stated or presumed to be primary, of lymphoid, haematopoietic and related tissue) | 2966 | 1000 |
| | D00-09 (In situ neoplasms) | 74 | 14 |
| | D10-36 (Benign neoplasms) | 1863 | 490 |
| | D37-48 (Neoplasms of uncertain or unknown behaviour) | 6691 | 2149 |

For each patient, a dataset (number of items × number of time steps) was created using test items, sex, height, weight, and vital signs. The preprocessing of each item followed the method outlined in Table 2. Detailed laboratory characteristics, including classification of abnormally low and abnormally high values for each test value, are provided in the S1 Appendix. The data were arranged chronologically based on the recording date and treated as time-series data. Of these, 4,668 patients whose deaths were confirmed from electronic medical records were considered deceased, and the remaining 8,027 patients whose deaths were not confirmed were considered survivors. Mortality information was excluded from the input of the deep state-space model because it was used only to interpret the results. The MinMax method normalizes the data points by rescaling its range to have a minimum value of 0 and a maximum value of 1. Missing data were accounted for in the deep state-space model using masked data, indicating the presence of missing data. Although the ratio of missing values in the entire dataset was 59.23%, in the deep state-space model described in the next section, these missing values can be naturally handled by masking them when calculating the loss function, except for the initial values. To verify the robustness of our model against high missing data rates, we performed a validation study using synthetic datasets, which is detailed in S3 Appendix. This ratio of missing values is calculated from the proportion of missing values for all variables, individuals, and time steps. The zeroth-order spline method performs interpolation of missing values using a constant function. The distribution of the number of time steps per patient record is shown in S1 Appendix. The maximum number of time steps for the bottom 90% of patients in the time-step count was 238, and the most recent record of 238 steps was retained for patients with a high number of time steps.

## Deep state-space model

The latent state was estimated using a deep state-space model [13], which is a deep-learning state-space model that can estimate the latent state of a time-series by inputting time-series observables, as shown in Fig 1a. The layers and hyperparameters of the deep state-space model, optimized using a validation set, are described in S2 Appendix. The final model was applied to the entire dataset for latent state estimation.

In the deep state-space model, a neural network was used to represent a probabilistic state-space model, and the parameters of the three neural networks were learned from the observed data through variational inference. The state-space model here considers the observation $x_t$ from the state $z_t$ at each time t ($t = 1, 2, \ldots, T$) and represents the transformation from state to observation as $p_\theta(x_t|z_t)$ and the state transition as $p_\theta(z_t|z_{t-1})$, where these probabilities are represented by multi-layer perceptron neural networks with parameters $\theta$. These probability distributions were represented using normal distributions parameterized using time-independent neural networks. These neural networks were trained using variational inference from the unsupervised observed data only.

**Table 2. Overview of the input features used for the model. This table details the four feature categories, the methods for data selection and preprocessing (such as normalization and missing value imputation), and the final dimensionality of each category.**

| Feature category | Data to be used | Preprocessing methods | Number of items |
|---|---|---|---|
| Height and weight | Use daily mode | Normalization: MinMax method<br>Missing value interpolation: Zero-order spline method (piecewise constant function) | 2 |
| Gender | Male: 1, Female: 0 | Enter the same gender for all time steps | 1 |
| Vitals | Use the daily mode of body temperature, pulse, max., and min. blood pressure | Normalization: MinMax method<br>Missing value interpolation: Zero-order spline method (piecewise constant function) | 4 |
| Laboratory tests | Use 50 items with the most abnormal values, except for correlation > 0.7<br>Abnormal high value: 1, Abnormal low value: 0, Normal value: 0.5 | Normalization: MinMax method<br>Missing value interpolation: Zero-order spline method (piecewise constant function) | 50 |

Variational inference introduces a new neural network-represented variational distribution $q_\phi$ for learning and learning the parameters of all neural networks by the gradient method by maximizing the lower bound of the $log\ p_\theta(x)$, which is the log-likelihood that is peripheral for the states derived as follows

$$log\ p_\theta(x) \geq \sum_{t=1}^{T} E_{q_\phi(z_t|x)} [log p_\theta(x_t|z_t)] - KL(q_\phi(z_1|x) \mid p_0(z_1))$$

$$- \sum_{t=2}^{T} E_{q_\phi(z_{t-1}|x)} [KL(q_\phi(z_t|z_{t-1},x) \mid p_\theta(z_t|z_{t-1}))]$$

(1)

where $KL$ represents the KL divergence, which is defined as $KL(q(z) \mid p(z)) = \int_Z q(z) log \frac{q(z)}{p(z)}\ dz$. The prior distribution $p_0(z_1)$of the initial condition was assumed to be normally distributed with a mean 0 variance of 1. The first term represents the term related to the reproduction of observations, the second term represents the regularization term owing to the prior of the initial distribution, and the third term represents the term related to the consistency of state transitions. Once these neural networks can be trained (i.e., maximize the right-hand side), the states can be estimated using $q_\phi(z_t|x)$, which approximates the posterior distribution.

In this study, the model was trained on a time-series of observations $x_t$, where each observation consists of a 57-dimensional laboratory test value (Table 2) and the time interval until the next measurement. In the following, we analyze the clustering and transitions using the behavior of the latent state $z_t$ at each time.

In this model, missing data can be taken into account during learning by sampling from the observation probability $p_\theta(x_t|z_t)$ and completing the missing values during learning, making it possible to learn while appropriately taking into account the presence of missing values. Meanwhile, to use this probability, a estimated state $z_t$ needs to be given, and this $z_t$ is estimated from the observations by the variational distribution $q$, so in the first step of learning, we use values filled in using traditional missing value imputation in Table 2, and from then on this model uses their own estimated values for missing values.

The code of this deep state-space model is available from https://github.com/clinfo/DeepKF.

## Visualization of latent states

To facilitate interpretation, the latent states represented in multiple dimensions were stratified using k-means, and dimension reduction using the UMAP technique was employed to visualize them in 2D. In Fig 2a, the endpoints of the latent states over time are plotted for deceased patients (red) and surviving patients (blue). Additionally, in Fig 2b, patient state stratification was conducted, and the latent states are plotted in clusters I (red), II (yellow), and III (green). The number of endpoints for deceased and surviving patients in each cluster is summarized. Furthermore, in Fig 2c, to visualize the temporal state transitions of patients, an example of the state transitions for deceased and surviving patients is illustrated. All latent states (blue) for all patients across all time-series are plotted; latent states that change over time are represented by colors that change from white to red and red to black, and color bars indicate the progression of days from the endpoint.

For the dimension reduction of the latent states represented in multiple dimensions to 2D, UMAP was used. The parameter selection for UMAP was as follows: first, the latent states were downsampled to 1/10 of the patient count, and trials were conducted with UMAP parameters such as the neighborhood size for probability density estimation of the data (n={15, 30, 50, 100}), a minimum distance between points in the embedded space (min-dist=0.1), and the number of dimensions after dimension reduction (d=2). As the choice of n did not significantly affect the visualization results in this trial, n=15 was selected. A sensitivity analysis demonstrating the stability of the clustering structure across different UMAP hyperparameters is provided in S4 Appendix. As the initial steps in patient time-series tend to involve challenging latent state estimation owing to data variability, visualization was performed, excluding the first three steps.

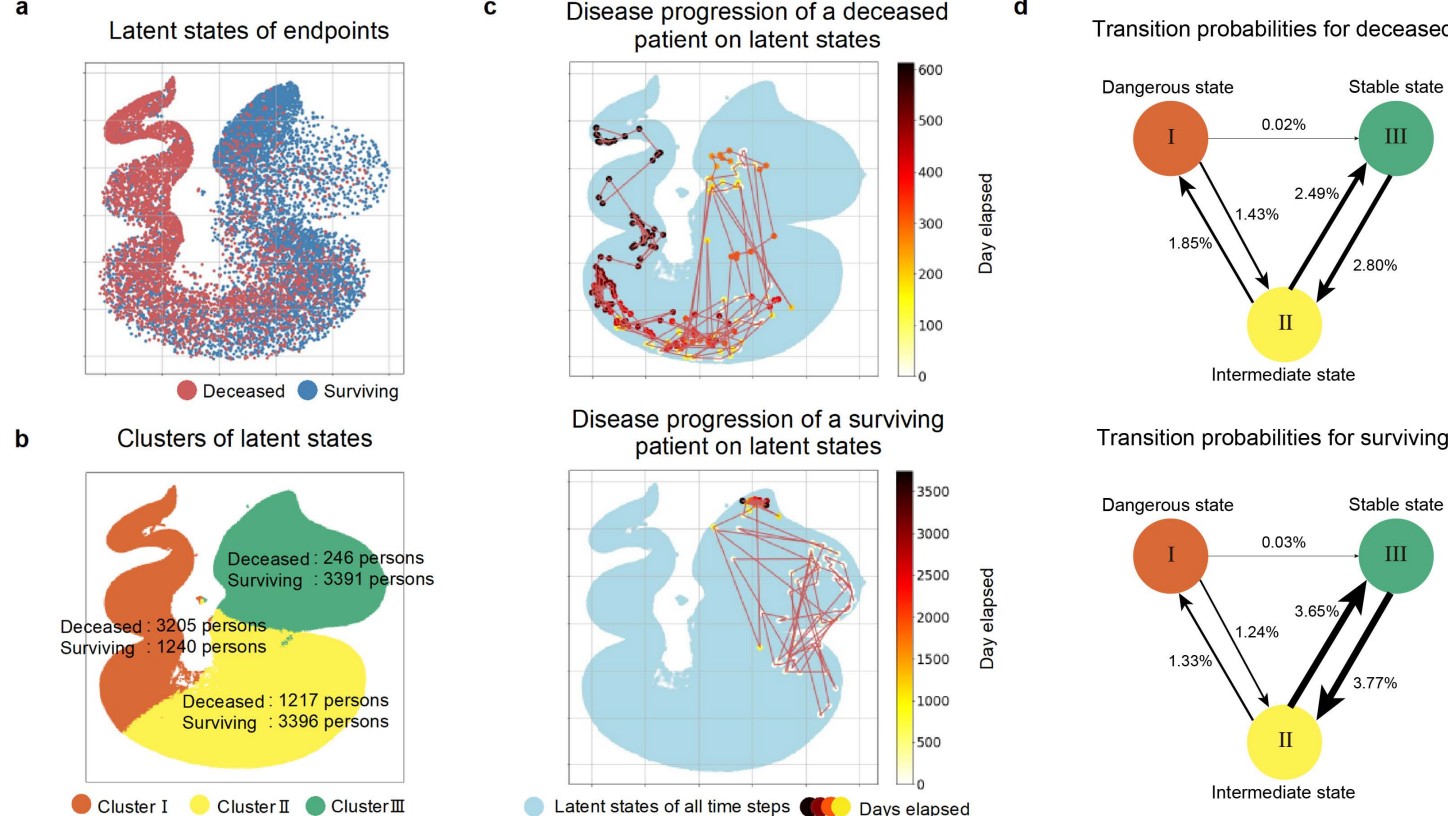

**Fig 2.** **(a) The differences in the distribution of endpoints over the time-series of latent states for deceased patients (red) and surviving patients (blue) are shown. (b)** The results of patient stratification and the number of endpoints for deceased and surviving patients in each cluster are shown. Cluster I (red), Cluster II (yellow), and Cluster III (green) correspond to the dangerous state, the intermediate state, and the stable state, respectively. **(c)** The state transitions for deceased and surviving patients are shown as an example. The blue plots represent the latent states of patients across all time-series, the color bar indicates the number of days elapsed from the endpoint, and the plots change from white to red, and from red to black over time. The example state transitions for deceased patients show transitions from cluster III to II to I, while the example for surviving patients shows transitions back and forth between clusters III and II. **(d)** The transition probabilities between the three clusters for deceased and surviving patients. The thickness of the lines between the clusters represents the magnitude of the probability.

Before performing dimension reduction using UMAP, k-means clustering was used to stratify patient states from latent states. For the number of clusters, a clustering evaluation metric called the silhouette score was high, and a cluster count of three with a low variance in the total sample count was chosen. Furthermore, as indicated in the section "Application of deep-state-space analysis framework to EHR," the death flag obtained from the EHR was used to identify the importance of latent states for patients immediately before death. Consequently, clusters whose endpoints in the time-series of deceased patients were concentrated were identified as in the dangerous state, clusters where endpoints in the time-series of surviving patients were concentrated in the stable state, and clusters where endpoints for both deceased and surviving patients were mixed in the intermediate state. Each cluster was then analyzed.

To investigate state transitions over time for patients, we considered transitions between clusters as Markov chains and calculated their transition probabilities for all patients, dead patients, and surviving patients. Specifically, we counted the number of transitions between clusters in each group and compiled the frequencies into tables.

## Extract temporal risk factors during state transitions

We investigated the characteristic test items for each cluster obtained by clustering the latent states and extracting the important test items during latent state transitions. Specifically, we created three datasets $D_I$, $D_{II}$, $D_{III}$, by associating EHR test data with cluster labels I, II, and III through the latent state points of each patient at each time point and grouping them by clusters. Using these datasets, we examined the relationship between the distributions of the test items represented by abnormally low, normal, and abnormally high values using the Wasserstein distance [16] as a measure. Wasserstein distance, denoted as $Wp(q_a, q_b)$ $(a, b \in \{I, II, III\})$ $Wp(q_1, q_2) =$, was used to compare the two empirical distributions $(q_a, q_b)$ of the three datasets $(D_I, D_{II}, D_{III})$ for each test item represented by abnormal low values, normal values, and abnormal high values. In our setting, we can compute $Wp(q_a, q_b) = \int |x_a - x_b| \, d\pi(x_a, x_b)$, where $\Gamma(q_a, q_b)$ is the set of all possible joint distributions whose marginal distributions are $q_a$ and $q_b$. We defined the sum of differences in distributions between each pair of clusters, W, as shown in the following equation (2):

$$W = Wp(q_I, q_{II}) + Wp(q_{II}, q_{III}) + Wp(q_{III}, q_I) \tag{2}$$

Test items with large W values are likely to vary significantly between clusters and can be considered characteristic items for each cluster. Here, we focused on the top ten test items with large W values and examined the proportions of abnormally high, normal, and abnormally low values in each cluster. We represent these proportions as bubble plots, with the size of the bubbles indicating the percentage of abnormal values. In Fig 3a, we investigated the temporal risk factors during state transitions in all cancer patients. Additionally, we treated test values within the institutional reference range as normal, values higher than the reference range as abnormally high, and values lower than the reference range as abnormally low. Furthermore, we focused on the top two items with large W values, hemoglobin (HGB) and hematocrit (HCT), and examined the distribution of abnormal values for each drug-administered cancer patient and each cluster. The results are presented in a stacked bar graph in Fig 3b. Test items with a high frequency of abnormal values are considered crucial factors during state transitions.

## Extract temporal risk factors during drug-dependent state transitions

To investigate drug-dependent temporal risk factors, we utilized the latent states of patients with cancer receiving each of eight drugs (Nivolumab, Trastuzumab, Cisplatin, Bicalutamide, Imatinib, Osimertinib, Afatinib, Erlotinib). These drugs were selected based on the highest number of patients prescribed at our institution to ensure a sufficient sample size for analysis (the specific number of patients administered each drug is detailed in S1 Table. 2). Furthermore, the analysis focused on 50 clinical test items specifically selected for their high frequency of abnormal values, representing parameters most likely to reflect significant physiological fluctuations during treatment. Following a methodology like "Extract Temporal risk factors during state transitions," we examined drug-dependent temporal risk factors during state transitions. In S1 Fig. and Fig 1, we investigate the temporal risk factors during state transitions for cancer patients receiving each anticancer drug, representing the proportion of abnormal values with a bubble plot. Here, we focused on lymphocytes and segmented neutrophils, which are specific test items for Nivolumab and Osimertinib and were not observed in the analysis of all patients with cancer. For each of these test items, we examined the distribution of abnormal values for each drug-administered patient with cancer and each cluster and presented the results in a stacked bar graph (Fig 3c).

## Comparison methods for latent state estimation

PCA, VAE, and linear state-space models were used as comparative methods for latent state estimation using deep state-space models. Specifically, PCA was employed as a simple linear dimensionality reduction method without considering time-series information, while VAE was used as a nonlinear dimensionality reduction method that also does not consider time-series information. The linear state-space model was used as a baseline method for time-series modeling.

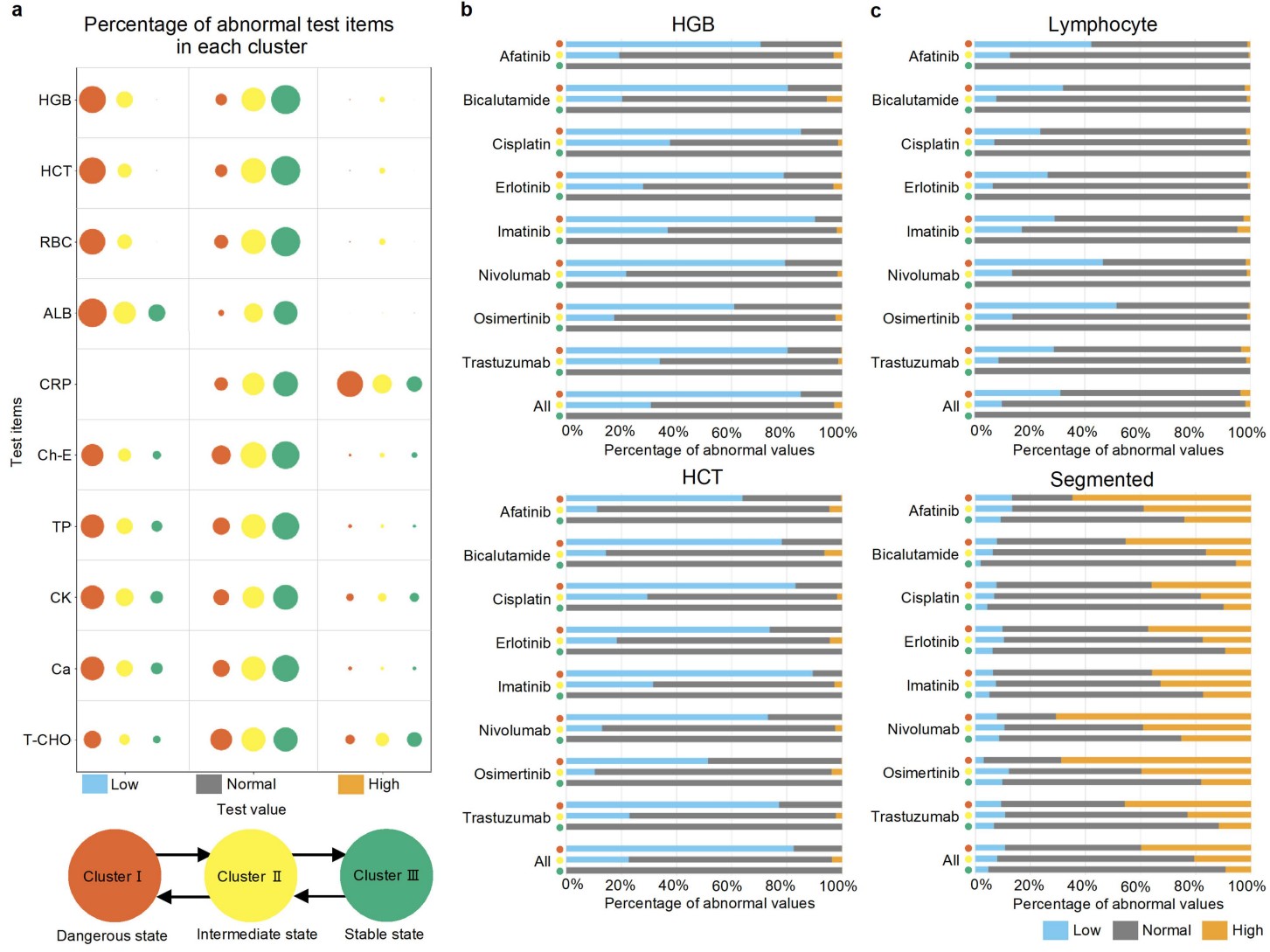

**Fig 3. (a) A bubble plot depicting the percentage of outliers for the top 10 items with a large difference in distribution (Wasserstein distance) between clusters.** The size of the bubble indicates the percentage of abnormally low, normal, and abnormally high values in each cluster. The x-axis indicates whether test values are abnormally low, normal, or abnormally high; the y-axis corresponds to the type of test item, and the color corresponds to the cluster. The items are arranged in descending order based on the magnitude of the difference in distribution between clusters. **(b)** A stacked bar graph illustrates the percentage of abnormal values for the top 2 items, HGB and HCT, with a difference in distribution between clusters. The graph shows the percentage of abnormal values for HGB and HCT in each drug-administered cancer patient and each cluster. In all patients with cancer receiving chemotherapy, there is a higher percentage of abnormally low values for HGB and HCT in Cluster I, which is a dangerous state. **(c)** For each patient with cancer receiving drug treatment, we examined the characteristic differences in the distribution between clusters for lymphocytes and segmented neutrophils values. The graph illustrates the percentage of abnormal values for each drug-administered cancer patient in each cluster. In patients treated with Afatinib, Nivolumab, and Osimertinib, there is a higher proportion of abnormally low values for lymphocytes and abnormally high values for segmented neutrophils in Cluster I, which is a dangerous state.

The VAE consists of two neural networks: an encoder and a decoder. The encoder has three fully connected layers, and the decoder has two fully connected layers. The values of epoch = 50 and learning rate = {0.001, 0.005, 0.01} were tested as hyperparameters, and the parameter learning rate = 0.01 was selected because it captured the mortality and survival rates most qualitatively. The linear state-space model is a linearized version of the neural network that constitutes a deep

state-space model. The 8-dimensional latent states obtained by applying the dimensionality reduction method under the same conditions as the latent state estimation using the deep state-space model were visualized in two dimensions using UMAP. We compared whether the latent states obtained by each dimensionality reduction method adequately captured the patient states and patient state transitions over time. Detailed configurations, network architectures, and training hyperparameters for these baseline models are provided in S2 Appendix.

## Results

### Application of deep-state-space analysis framework to EHR

To evaluate the clinical interpretability of the estimated latent state-space within the proposed deep state-space analysis framework, the prognosis of patients with cancer was used as a clinical indicator. Specifically, we focused on 12,695 patients with cancer who underwent chemotherapy at Kyoto University Hospital between 2006 and 2018. We categorized the patients into those who were alive until the last EHR data acquisition point in 2018 and those who had already passed away. The evaluation focused on two aspects: first, assessing whether latent states accurately capture the favorable or unfavorable conditions of patient prognosis, allowing for a clinically meaningful interpretation of the patients, and second, confirming the spatiotemporal state transitions within the latent space up to the death of the patient, aiming to link these transitions to the identification of risk factors.

First, we confirmed whether the estimated latent states appropriately captured good or poor prognoses of patients with cancer. Fig 2a–2c depict the latent state-space estimated using the deep state-space model and reduced to two dimensions using UMAP. Fig 2a shows the last EHR data acquisition point for the deceased and surviving patients. Fig 2b shows the results of clustering latent states across all times for all patients using k-means and the number of deceased and surviving patient endpoints within each cluster. By examining the distribution of endpoint counts for deceased and surviving patients in Clusters I (red), II (yellow), and III (green), Cluster I appeared to have a concentration of endpoint events for deceased patients, suggesting a high risk of death and a dangerous state. In contrast, Cluster III showed a concentration of endpoint events for surviving patients, indicating a relatively stable state distant from death. Cluster II had a mixed distribution of endpoint events for deceased and surviving patients, suggesting an intermediate state. Therefore, we defined Cluster I as the "dangerous state," Cluster II as the "intermediate state," and Cluster III as the "stable state." Thus, the estimated latent state-space clearly separates the endpoint events for deceased and surviving patients, confirming that the latent states appropriately capture the clinical conditions of the patients.

Next, we confirmed the spatiotemporal state transitions within the latent space up to the death of the patient to identify the temporal risk factors. To visually depict the temporal transitions of the latent states of individual patients, Fig 2c shows examples of representative state transitions for the deceased and surviving patients. The visual representation indicated that patient states gradually changed over time, with deceased patients transitioning to a dangerous state and surviving patients tending to remain in a stable state. Furthermore, to assess the validity of the temporal transitions in the estimated latent states and three clusters, a Markov chain model was used to calculate the transition probabilities within and between clusters for the deceased and surviving patients. The probability of staying within a cluster was high for many patients, whereas the probability of transitioning between clusters was low. Fig 2d shows the transition probabilities between clusters for the deceased and surviving patients. The frequency of transitions from Cluster I to II and from Cluster II to III was higher than that from Cluster I to III. Thus, the primary patterns were transitions between Clusters I and II and Clusters II and III. Additionally, for the deceased patients, there were more transitions between Clusters I and II, whereas for the surviving patients, there were more transitions between Clusters II and III. Details of the transition frequencies between clusters are provided in S1 Table. This quantitative analysis confirmed that deceased patients tended to transition to a dangerous state, whereas surviving patients tended to remain stable.

## Identification of temporal risk factors

To discover temporal risk factors using the latent states obtained through this framework, we narrowed down the critical test items during the latent state transitions. Fig 3a presents the results of calculating the proportions of abnormally low, abnormally high, and normal values for each test item in each cluster for the top 10 test items with significant differences in distribution between clusters, as represented by the Wasserstein distance. When comparing the dangerous state of Cluster I with the intermediate state of Cluster II, certain test items, such as hemoglobin (HGB), hematocrit (HCT), and red blood cell count (RBC), showed a higher proportion of abnormally low values. Focusing on the top two test items with significant differences in distribution between clusters, HGB and HCT, we investigated the proportions of abnormal and normal values for these test items in Clusters I, II, and III of patients with cancer treated with different anticancer drugs, as shown in Fig 3b. From Fig 3b, it is evident that low HGB and HCT values were significant temporal risk factors common to all patients with cancer.

Furthermore, to extract important temporal risk factors during state transitions from the latent states of patients with cancer treated with different anticancer drugs, we focused on the eight drugs listed in Table 3 and evaluated the prognostic risk for patients dependent on the administered drugs. S1 Fig. shows that test items, such as lymphocytes and segmented neutrophils, were characteristic of cancer patients treated with Nivolumab and Osimertinib. Focusing on these two test items, we investigated the proportions of abnormal and normal values for these test items in Clusters I, II, and III of patients with cancer treated with different anticancer drugs (Fig 3c). As shown in Fig 3c, it became evident that low lymphocyte counts and high segmented neutrophil counts were characteristic temporal risk factors in patients with cancer treated with Afatinib, Nivolumab, and Osimertinib.

## Comparison with the conventional methods

For comparison with the deep state-space model, we present the estimation results of latent states using three different comparison methods: Principal Component Analysis (PCA), Variational Autoencoder (VAE), and a linearized version of our model referred to as the "Linear state-space model" in Fig 4. It should be noted that the VAE is also a deep state-space model without temporal transitions.

The top panel of Fig 4a shows the latent states at the endpoints of the time-series, where the red and blue plots represent the deceased and surviving patients, respectively. For the PCA and VAE, there was little distinction between the red and blue plots, whereas, for the linear state-space model, there was only a slight separation. In contrast, the deep state-space model clearly separates the red and blue plots spatially.

The bottom panel of Fig 4b illustrates the transition in the latent states over time for individual patients. Although both the linear and deep state-space models confirmed the spatiotemporal continuity of latent states, PCA and VAE, which are

**Table 3. The eight anticancer drugs selected for the analysis of temporal risk factors. The table lists each drug and its corresponding drug classification.**

| Drug name | Classification |
| --- | --- |
| Nivolumab | Immune checkpoint inhibitors (anti-PD-1 antibodies) |
| Trastuzumab | Molecular targeted drugs (anti-HER2 antibodies) |
| Cisplatin | Platinum drugs |
| Bicalutamide | Hormone therapy drugs |
| Imatinib | Molecularly targeted drugs (tyrosine kinase inhibitors) |
| Osimertinib | Molecularly targeted drugs (tyrosine kinase inhibitors) |
| Afatinib | Molecularly targeted drugs (tyrosine kinase inhibitors) |
| Erlotinib | Molecularly targeted drugs (tyrosine kinase inhibitors) |

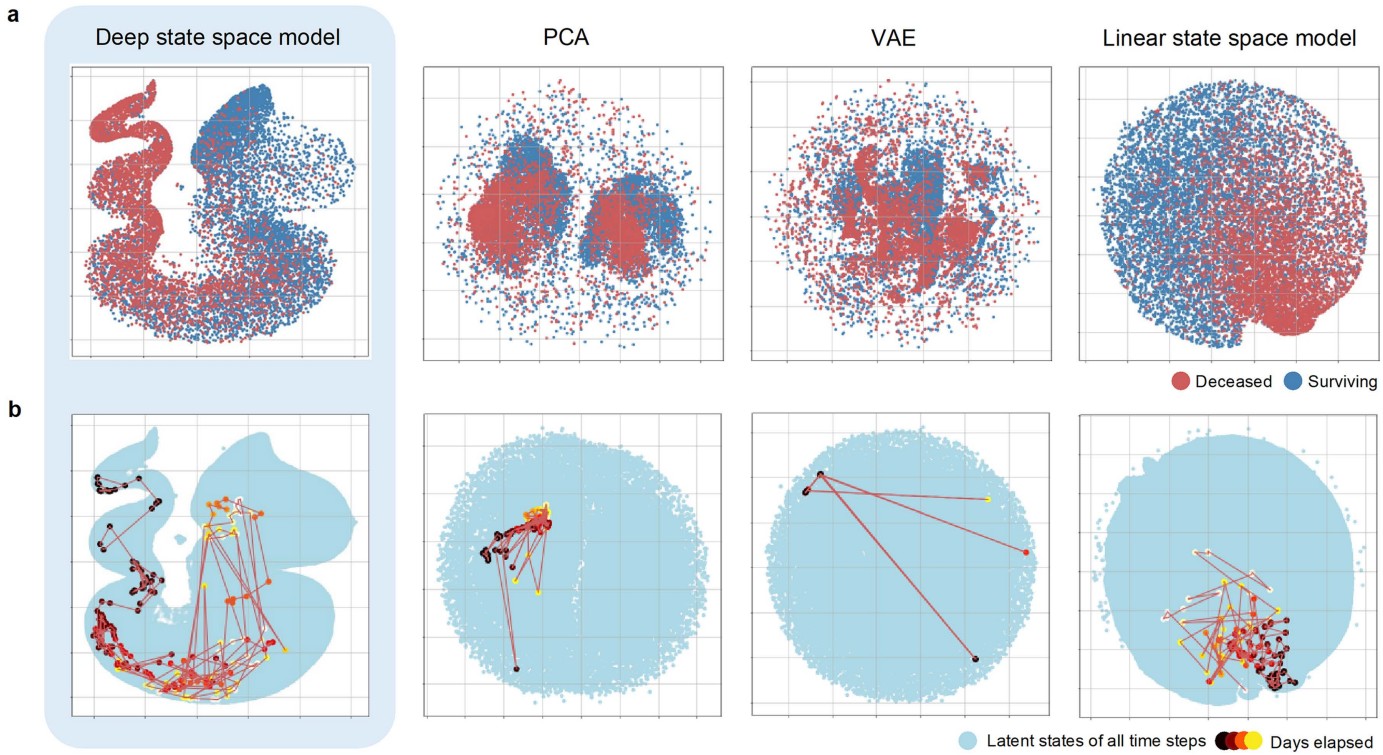

**Fig 4. (a) The differences in the distribution of endpoints over time for deceased patients (red) and surviving patients (blue) in the time-series of latent states obtained by the proposed method (deep state-space model) and each comparison method (PCA, VAE, linear state-space model) is shown. (b)** This illustrates the state transition of a deceased patient as an example. The blue plots represent the latent states of all patients across all time-series and the plots change from white to red and then to black as time progresses.

latent state estimation methods that do not consider temporal transitions, were unable to verify the spatiotemporal continuity in state transitions. Thus, the deep state-space model successfully captured the temporal evolution of the prognosis of patients with cancer, which conventional methods cannot achieve.

## Discussion

To confirm the clinical validity of the latent state, two points were evaluated: whether the latent state captures the condition of the patient, and whether it captures the spatiotemporal continuity leading to death. As shown in Fig 2a and 2b, the latent states obtained using the deep state-space model successfully captured both patient survival and death. Additionally, as depicted in Fig 2c and 2d, the model captured the progression of the state of the deceased patients towards a dangerous state over time.

Our developed framework was able to accurately capture the patient's condition and its temporal changes, a point that is evident even when compared with existing latent state estimation methods (see Fig 4a and 4b). For example, a non-linear model that does not consider time-series information (VAE) could not clearly separate the states of surviving and deceased patients, nor could it capture the continuous changes in state over time. Similarly, while a simple time-series model (linear state-space model) captured some continuity, it failed to achieve the clear separation of final states between deceased and surviving patients that our model's "non-linearity" provides. Previous research has suggested that models might implicitly learn temporal continuity even without being explicitly designed for it [17,18]. However, our results demonstrate that when dealing with complex EHR data, directly modeling the passage of time is crucial for capturing clinically

meaningful patient states, such as prognosis. In conclusion, the major advantage of our model is that by learning both "temporal transitions" and "non-linearity," it does not merely reduce data dimensions but estimates highly interpretable latent states that allow clinicians to intuitively understand a patient's prognosis.

Furthermore, in this study, we focused on eight anticancer drugs and identified the characteristic temporal risk factors associated with each drug administration in patients with cancer.

Based on the results from the "Identification of Temporal Risk Factors" section, the eight targeted anticancer drugs focused on in this study (Table 3) were classified into immune checkpoint inhibitors (anti-PD-1 drugs), molecular targeted drugs (anti-HER2 drugs, tyrosine kinase inhibitors), platinum agents, and hormone therapy drugs. Drug-dependent temporal risk factors were identified, particularly for immune checkpoint inhibitors and tyrosine kinase inhibitors, for which immune system abnormalities were highlighted. Abnormally low lymphocyte counts and high segmented neutrophil values were significant temporal risk factors for patients receiving immune checkpoint inhibitors, such as Nivolumab, and tyrosine kinase inhibitors, such as Osimertinib and Afatinib. The observed abnormalities in the immune cell system during state transitions in patients receiving Nivolumab are consistent with previous reports, where immune-related adverse events (irAEs) were identified as adverse prognostic factors [19]. Furthermore, the neutrophil-to-lymphocyte ratio (NLR) has been reported to be an important adverse prognostic factor for tyrosine kinase inhibitors [20], with particular emphasis on Osimertinib- and Afatinib-treated patients with cancer as potentially crucial adverse prognostic factors.

In this study, we successfully modeled temporal changes through latent state estimation, captured clinical validity, and identified temporal risk factors. A challenge thus far has been the difficulty in estimating explainable disease progression from observed patient test data owing to the black-box nature of deep learning. However, by focusing on and interpreting latent states, along with combining visualization and stratification, we achieved success in capturing clinical validity and temporal risk factors in latent state estimation modeling of temporal changes, constructing a clinically understandable framework.

This study has a few limitations. A limitation of this study is its retrospective and single-center design, which makes it challenging to establish specific factors and causal relationships. In this study, anemia was estimated as a temporal risk factor in patients with cancer. Although potential causes of anemia in cancer patients undergoing chemotherapy include myelosuppression, bleeding, bone marrow infiltration, treatment-related factors, and malnutrition due to cancer-induced chronic inflammation, identifying the background of the anemia is still difficult. To address this issue, it is essential to formulate hypotheses based on the results of this framework, and more detailed experiments such as prospective studies are needed to further elucidate relationships. Additionally, death flags are not necessarily based on patients who died due to cancer, as the data includes patients who died for other reasons.

Furthermore, while this study analyzed patients with various cancer types collectively to establish a generalized framework, we acknowledge the clinical interest in subtype-specific analyses. However, stratifying the dataset into single cancer types resulted in insufficient sample sizes for many groups, which leads to unstable estimations given the data-intensive nature of deep learning models. Therefore, detailed analyses for specific cancer subtypes were not feasible in this study, and we consider this a challenge to be addressed in future research with larger datasets.

Another limitation is that the state transitions are modeled using deep neural networks, and the approach involves learning from EHR test data. Therefore, the detailed mechanisms of the state transitions are not fully understood. To enhance the understanding, further investigation into the background information of abnormal values detected by this framework is needed, to elucidate the key factors and fundamental mechanisms contributing to transitions into dangerous states.

Additionally, in recent chemotherapies for patients with cancer, combination therapy involving multiple drugs has become common. Identifying temporal risk factors during state transitions for combinations of multiple drugs is challenging because of the significant side effects and economic burden on patients. Therefore, future research should focus on more detailed studies to identify important test items during state transitions for different anticancer drugs and combinations of multiple drugs that were not analyzed in this study.

## Conclusion

We developed a deep state-space analysis framework for estimating and visualizing latent states from EHR using a deep state-space model. This framework was applied to the EHR of patients undergoing chemotherapy to enable clinically interpretable and temporal risk factor-specific time-series analysis. This interpretation reveals the three latent states and their relationships over time. These results identified potentially important temporal risk factors in cancer patients treated with anticancer drugs. Furthermore, the latent states obtained were used for the visualization and interpretation of cancer patients treated with specific anticancer drugs to obtain test items that are characteristic of cancer patients treated with specific anticancer drugs and important during state transitions. We believe that this approach will be useful as an analytical framework for time-series EHR data, and that it may be possible to incorporate our framework into EHR analysis tools in the future, which will hopefully enable us to capture new dynamic patient characteristics.

In the future, leveraging this framework for chronic diseases that require long-term monitoring, not limited to cancer, holds the potential to advance patient health management using latent states, optimize treatment strategies, such as early treatment modification and prognosis assessment, and contribute to further advancements in healthcare.

## Supporting information

**S1 Appendix. Overview of the analysis dataset.** This appendix provides an overview of the analysis dataset, including its composition and key characteristics.
(DOCX)

**S2 Appendix. Model architecture and hyperparameters.** This appendix details the model architecture and the hyperparameters used in the study.
(DOCX)

**S1 Table. Transition probabilities between clusters.** This table shows the probability of transitioning between each pair of clusters identified in this study.
(DOCX)

**S1 Fig. Percentage of abnormal test items in each cluster.** This figure shows the percentage of test items classified as abnormal in each cluster, highlighting differences in abnormal result rates among clusters.
(TIF)

**S3 Appendix. Robustness of DeepSSM to missing data rates.** This appendix validates the robustness of a Deep State-Space Model against missing data in NCAR and MNAR scenarios using Langevin dynamics-based synthetic simulations.
(PDF)

**S4 Appendix. Sensitivity analysis of UMAP hyperparameters.** This sensitivity analysis confirms that the patient clustering results are robust to changes in UMAP hyperparameters.
(PDF)

## Author contributions

**Conceptualization:** Yuji Okamoto, Aya Nakamura, Ryosuke Kojima.

**Data curation:** Yuji Okamoto, Aya Nakamura, Ryosuke Kojima, Eiichiro Uchino, Yohei Mineharu, Yohei Harada, Mayumi Kamada.

**Formal analysis:** Yuji Okamoto, Aya Nakamura, Ryosuke Kojima.

**Investigation:** Yuji Okamoto, Aya Nakamura, Ryosuke Kojima, Eiichiro Uchino, Yohei Mineharu, Yohei Harada, Mayumi Kamada, Minoru Sakuragi.

**Methodology:** Yuji Okamoto, Aya Nakamura, Ryosuke Kojima, Eiichiro Uchino, Yohei Mineharu, Yohei Harada, Mayumi Kamada.

**Project administration:** Manabu Muto, Motoko Yanagita, Yasushi Okuno.

**Resources:** Yuji Okamoto, Aya Nakamura, Ryosuke Kojima.

**Software:** Aya Nakamura, Ryosuke Kojima.

**Supervision:** Manabu Muto, Motoko Yanagita, Yasushi Okuno.

**Validation:** Yuji Okamoto, Aya Nakamura, Ryosuke Kojima, Eiichiro Uchino, Yohei Mineharu, Yohei Harada, Mayumi Kamada, Minoru Sakuragi.

**Visualization:** Yuji Okamoto, Aya Nakamura, Ryosuke Kojima, Eiichiro Uchino, Yohei Mineharu, Yohei Harada, Mayumi Kamada, Minoru Sakuragi.

**Writing – original draft:** Aya Nakamura, Ryosuke Kojima.

**Writing – review & editing:** Yuji Okamoto, Eiichiro Uchino, Yohei Mineharu, Yohei Harada, Minoru Sakuragi, Manabu Muto, Motoko Yanagita, Yasushi Okuno.

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
