## [Decision Letter · Decision Letter 0]

16 May 2025

Dear Dr. Okuno,

Thank you for submitting your manuscript to PLOS ONE. After careful consideration, we feel that it has merit but does not fully meet PLOS ONE’s publication criteria as it currently stands. Therefore, we invite you to submit a revised version of the manuscript that addresses the points raised during the review process.

We look forward to receiving your revised manuscript.

Kind regards,

Zeheng Wang

Academic Editor

PLOS ONE

Journal Requirements:

[This work was supported by JST, Center of Innovation Program (JPMJCE1302).].

[This work was supported by JST, Center of Innovation Program (JPMJCE1302). This research was supported by JST Moonshot R\&D Grant Number JPMJMS2021 and JPMJMS2024.]

[This work was supported by JST, Center of Innovation Program (JPMJCE1302).].

6. In the online submission form, you indicated that [Data cannot be shared publicly because of patient privacy in electronic medical records. Data are available from the corresponding author and Kyoto University Graduate School and Faculty of Medicine, Ethics Committee via email (ethcom@kuhp.kyoto-u.ac.jp) or telephone (+81-75-753-4680) for researchers who meet the criteria for access to confidential data.].

7. We notice that your supplementary figures are uploaded with the file type 'Figure'. Please amend the file type to 'Supporting Information'. Please ensure that each Supporting Information file has a legend listed in the manuscript after the references list.

Additional Editor Comments:

The manuscript presents a potentially valuable contribution to the field of AI for healthcare, particularly in its practical relevance and application-driven insights. Several reviewers acknowledged the importance of the problem being addressed and the promise of the proposed approach. However, significant concerns were raised regarding the technical presentation of the work. Specifically, the manuscript lacks the clarity, precision, and methodological rigor typically expected in AI-focused publications.

The current version suffers from a lack of domain-appropriate language and structure, which obscures the novelty and reproducibility of the methods. Multiple reviewers have recommended rejection due to these issues. Nevertheless, given the potential practical impact of the work, we have decided to offer the authors an opportunity to substantially revise and resubmit.

We strongly recommend that the authors seek assistance from a researcher or collaborator with expertise in AI or machine learning to thoroughly revise the manuscript, ensuring that the technical descriptions are accurate, the methods are transparent and reproducible, and the overall narrative meets the professional standards of the AI research community. A professional proofread focused on AI terminology and methodology is highly encouraged.

Only after a comprehensive revision addressing these concerns will the manuscript's suitability be reconsidered again for publication.

Reviewers' comments:

Reviewer's Responses to Questions

**Comments to the Author**

1. Is the manuscript technically sound, and do the data support the conclusions?

Reviewer #1: Yes

Reviewer #2: Partly

Reviewer #3: Yes

Reviewer #4: Yes

Reviewer #5: Yes

Reviewer #6: Partly

2. Has the statistical analysis been performed appropriately and rigorously?

Reviewer #1: I Don't Know

Reviewer #2: No

Reviewer #3: Yes

Reviewer #4: Yes

Reviewer #5: I Don't Know

Reviewer #6: No

3. Have the authors made all data underlying the findings in their manuscript fully available?

Reviewer #1: Yes

Reviewer #2: Yes

Reviewer #3: No

Reviewer #4: No

Reviewer #5: Yes

Reviewer #6: Yes

4. Is the manuscript presented in an intelligible fashion and written in standard English?

Reviewer #1: Yes

Reviewer #2: Yes

Reviewer #3: Yes

Reviewer #4: Yes

Reviewer #5: Yes

Reviewer #6: Yes

Reviewer #1: This study introduces a deep state-space analysis framework to estimate latent patient states from electronic health records (EHR) and identify temporal risk factors for cancer progression. Overall, this framework advances interpretable deep learning for EHR analysis, offering a promising tool for personalized prognosis and treatment optimization. Addressing limitations in data accessibility and clinical complexity could amplify its impact in healthcare.

Strengths:

Innovative Methodology: Combines deep state-space models with visualization and clustering to enhance interpretability of temporal EHR data.Clinical Relevance: Identifies actionable prognostic factors (e.g., anemia, immune markers) validated on a large cancer cohort.Scalability: Framework generalizes across heterogeneous EHR data, enabling long-term disease progression modeling.

Areas for Improvement in the Manuscript:

Methodological Clarity:Provide detailed explanations of missing data handling (e.g., masking implementation) and model hyperparameter optimization processes. Clarify how the deep state-space model balances computational efficiency with accuracy, especially for large-scale EHR data.

Experimental Design: Address potential selection bias from excluding patients with <50 time steps and discuss its impact on generalizability.Acknowledge temporal shifts in medical practices or EHR recording standards (2006–2018) and their influence on data consistency.

Result Interpretation:Elaborate on the causal relationship between identified risk factors (e.g., anemia, immune markers) and disease progression, beyond correlation. Analyze heterogeneity across cancer types (Table 2) to assess whether results vary by malignancy.

Reviewer #2: The authors propose a deep state-space analysis framework that uses sequential EHR data to model and visualize patients’ latent disease states over time, enabling clustering by disease severity and identification of poor prognostic factors. The overall idea of the manuscript is nice. However, I have the following suggestions:

1. It would be better to have more consistent and precise terminology. The use of scientific and analytical language throughout the paper should be strengthened to improve clarity and help readers better grasp the study’s focus.

2. The structure of the manuscript may need to be revised. The Results and Discussion sections currently precede the Methods section. Also, part of the descriptions that belong in the Methods section can be considered relocated as part of the Results.

3. The data has been categorized several times without providing the criteria or standards. These should be explicitly defined and explained in the relevant sections.

4. The existing methods and your proposed approach should be clearly distinguished. Proper citations must be provided for existing techniques, and the manuscript should clearly state what the methodological contributions of the current work are. What is new or innovative about your approach should be made transparent.

5. Although the manuscript aims to demonstrate that deep learning is well-suited to model complex EHR data, the Methods section lacks a clear explanation of how EHR data is actually utilized in the proposed model.

6. Several descriptions in the Methods section lack clarity and precision. Please ensure the methodological steps align with the original techniques, unless improvements are introduced, in which case the modifications should be clearly stated. Terminology and phrasing must remain consistent throughout the manuscript to avoid confusion. For example, in lines 345–346, how the parameters obtained should be described clearly without omitting essential steps.

7. Are the UMAP parameters default? If parameter adjustments were made, please specify which ones and discuss their impact on the results.

8. The association between EHR test data and cluster labels (I, II, III) should be clarified.

9. The meaning of “abnormally high,” “normal,” and “abnormally low” values in each cluster should be defined within the context of this study. How is “abnormal” defined?

10. All figures should clearly label the x- and y-axes, including what each axis represents. The manuscript should include a clear description of what the figures show and their significance. Hard to correlate with text.

11. The Results and Discussion sections need more in-depth analysis. Statistical evaluations should be included to support findings, and thorough interpretations of the figures and results are necessary to enhance scientific rigor.

Reviewer #3: I enjoyed this paper and commend the authors for employing a new method to reduce dimensionality and merge temporal states and observational data. I especially appreciated the attention given to explaining the data pre-processing steps and domain knowledge used in the interpretation and use of laboratory tests. The machine-learning steps are well documented in the extended methods and are robust and supported.

Major Revision

While this paper presents a novel approach that integrates both temporal states and clinical test inputs to predict patient prognoses, the discussion section would benefit from additional supporting information and clarification:

• Your models identified anemia as a major prognostic factor. However, anemia is already a well-documented adverse effect of many of the therapies you mention, and has been widely cited as a critical clinical indicator. Did your model uncover any previously unrecognized patterns or associations involving anemia that are not well-established in the existing literature?

• How can your methods be effectively communicated to clinicians or individuals without a strong background in machine learning? What strategies or model characteristics support the trustworthiness and transparency of your predictions?

• Can you elaborate on how your methods could be adapted into clinical tools or algorithms for real-world use? For example, how might this approach be integrated into clinical decision support systems?

• Although there are understandable limitations to sharing raw patient data, could you make de-identified or simulated datasets available to support reproducibility? Your GitHub repository appears to include sample data, but this isn’t clearly highlighted—consider improving its visibility or documentation.

Minor Revision

• In Figure 1, consider making the representation of “time states” more explicit. Since temporality is a key element of your method, it would be helpful to clearly delineate different timepoints in the disease course and data collection timeline.

• Line 106: Revise to simply say “patient prognoses”—this term encompasses both favorable and unfavorable outcomes.

• Please include a brief discussion of the limitation associated with assuming survival in the absence of a recorded death in the EHR. This introduces a form of data censoring. Although most patients in your study were followed for less than two years—limiting the impact of this assumption—it may be worth suggesting future studies consider follow-up dates or discharge summaries to better estimate survival. This may be a point of confusion with clinicians who regularly employ censoring in survival analyses.

• Please check language consistency and accuracy for all figures. I found a few typos.

Reviewer #4: In this article, the authors have proposed a deep state-space analysis framework to estimate the hidden status for patients based on their observed electronic health records (EHR) data in an unsupervised fashion, to cluster the disease severity from the estimated hidden status and to identify the leading risk factors/driven biomarkers associated with each defined stratum. By accounting for the nonlinearity and time-dependency features in their framework, the deep state-space analysis method shows higher interpretability for its results compared to the conventional methods (principal component analysis, variational autoencoder, and linearized version of the proposed method). Generally speaking, this manuscript has carefully investigated the usage of deep learning methods in EHR analysis and has introduced a novel deep learning framework showing successful application in cancer research. I have several humble comments which I hope that the authors could address to further enhance the clarity and reproducibility of their research.

Major comments:

(Lines 248-254) Under either retrospective or prospective design, I have no clear idea why this proposed framework can detect causal effects in observational study as it always suffers from unmeasured confounding. I hope the authors could explain this point in more details regarding the validity of this method in the context of causal inference.

(Lines 331-333) Could you please provide more details about how the missing rate was calculated and how it is distributed at individual-level and variable-level (i.e., per-individual missing rate and per-variable missing rate). Also, I hope the authors could further explain how your proposed framework can handle large proportions of missing values.

(Table 2) In this table, the authors summarize the demographic and disease-related information for the 12,695 cancer patients in their analysis. Could you please show why the summation of the sample size for each cancer subtype is smaller than the total sample size?

(Figure 1) In the “Visualization of latent states” section, the authors mention that K-means clustering is performed before the UMAP dimension reduction (see Lines 384-386). However, Figure 1 illustrates that the UMAP step is performed prior to the clustering of latent states. What is the appropriate order of these procedures?

(Figure 2d) Please provide more details on interpreting these transition probabilities. Moreover, we observe from this figure that (1) the difference between stable -> intermediate probability and intermediate -> stable probability is 0.31% for deceased individuals and 0.12% for surviving individuals, and (2) the dangerous -> intermediate probability is larger than intermediate -> dangerous probability for deceased individuals while the dangerous -> intermediate probability is smaller than intermediate -> dangerous probability for surviving individuals. These two observations seem indicating that the deceased group is more likely to move from dangerous state to intermediate state as well as from intermediate state to stable state compared to the surviving group – could you please explain the reasons behind these observations?

Minor comments:

(Lines 327-328) In the definition of death across this study, does it mean the death due to cancer specifically, or it can arise due to other competing risks?

(Line 346) Which three networks are referred to in the phrase “the parameters of the three neural networks”?

(Lines 356-357) In the formula (1), in q distribution, x is not indexed by t – does it mean observed x across all time points rather than at time t?

(Lines 362-363) The authors mention that the hidden states can be estimated using q distribution - which estimates/summary statistics are used here?

(Lines 367-368) What is the interpretation of the original latent states without downstream dimension reduction and clustering procedures, or they just have no interpretations?

(Line 439) This sentence is incomplete after “linear state-space model”.

(Lines 443-445) What do the authors mean by “the 8-dimensional latent states”? Do they correspond to the 8 drugs?

(Table 3) Please provide more details regarding what MinMax method and Zero-order spline method are and how you applied them in this study. Also, by saying max. and min. blood pressure, do they mean systolic and diastolic blood pressure?

Reviewer #5: Dr. Suad Ghaben

PONE-D-25-02277

A New Deep state-space Analysis Framework for Patient Latent State Estimation and Classification from EHR Time-Series Data.

Dear Authors

Please find my comments on your manuscript below. Your manuscript is an observational study that analyzed patients’ data and identified a framework for predicting the latent state of chronic disease based on the HER time-series data. Your results highlighted the potentially important temporal risk factors related to patients with cancer undergoing chemotherapy. The identified framework would help to deepen the understanding of disease progression and supports early treatment adjustments, prognostic evaluations, and the formulation of optimal long-term strategies. I commend the authors on completing this essential work.

I will start by commenting on the manuscript parts in order highlighting major and minor revisions.

Title:

Short, concise, but not reflective enough. Please add patients with cancer to the title, and no need to include the “New”.

Suggested title: The Deep state-space analysis framework for estimating the Latent state for patients with cancer undergoing chemotherapy by utilizing the HER time-series data.

Abstract:

• PLOS ONE use structured abstract. Add the headings: Introduction, Materials and methods, Results, Discussion, and amend accordingly.

• Line 23: use “identified” instead of “proposed”, as you already identified and validated the framework using the HER time-series data of 12,695 patients.

Introduction:

• Line 74: In this study, we applied this framework to the EHR of 12,695 patients. Which framework? be specific and identify the framework.

• Lines 78-81: advantages of the identified framework can be mentioned in the results, discussion and conclusion sections, rather than introduction. In the introduction, you focus on the need and rational for developing the framework. You may paraphrase this section and highlight the advantages of the identified framework in the results, discussion and conclusion sections.;

Methods:

• Lines 267- 447: Move the methods section ahead to the Results section and keep consistency of their subheadings.

• Lines 340- 364: it’s unclear whether the described “Deep state-space model” if from literature or the new model that was applied in this study. Cite where required to highlight literature, and paraphrase to highlight your work in this study.

• I suggest add a paragraph to highlight the former deep stat-space model and the new one. You may depict a figure\diagram for more clarity.,

Results:

• Written well. No comments.

• keep consistency of the methods and results sections.

Discussion:

• written well. The interpretation is rational.

Conclusion:

• written well. No comments.

General comments:

• The manuscript is lengthy; please revise for writing in a more concise language.

• The citation style is not aligned with PLOS ONE guidelines; change the citation style to Vancouver.

• The manuscript should be thoroughly revised and rearranged; move the methods section ahead to the results section and keep consistency of subheadings for both sections.

Reviewer #6: In this paper, Authors developed a new framework called the "Deep state-space analysis framework" to enable the clinical interpretation of patients and the identification of temporal risk factors in the latent state space by explicitly modeling the temporal changes of patient latent states. Authors used endpoints of latent state space as indicators for cancer patients’ prognosis outcomes. They estimated state transition probabilities and identified key risk factors that are distinguishable among the three states. They also compared the new framework to several other latent space embedding methods.

Overall, the manuscript is thorough, but some aspects of the statements lack quantitative analyses. Here are some of my comments and questions:

1. The proposed framework lacks proof of quantitative validation, i.e., what is the model’s performance in making predictions?

2. Based on the metadata, what is the state transition difference between different cancer types, gender, age group, etc.?

3. The comparison to other latent space embedding methods lacks quantitative results. UMAP is not a plausible means of comparison for clustering.

4. The states transition plots, e.g. Fig 2c and 4b, are difficult to follow, thus require clarification and better interpretation.

5. The bubble plot, e.g. Fig 3a lacks figure legends. What does the difference between different sizes of the dots imply? On what scale?

6. Can you find any results in the literature that supports your conclusions in the identified key risk factors, or vice versa?

**Do you want your identity to be public for this peer review?** For information about this choice, including consent withdrawal, please see our Privacy Policy

Reviewer #1: No

Reviewer #2: No

Reviewer #3: No

Reviewer #4: No

Reviewer #5: **Yes:** Suad Ghaben

Reviewer #6: No

---

## [Author Response · Author response to Decision Letter 1]

19 Jul 2025

Manuscript Title:

A Deep State-space Analysis Framework for Cancer Patient Latent State Estimation and Classification from EHR Time-Series Data

Response to the Editor

Thank you for your comment regarding the Funding Information.

As you suggested, we have moved this information from the main text to the cover letter.

Response to Reviewer Comments

Thank you very much for your constructive and detailed feedback on our manuscript. Based on the reviewers' comments, we have made the following substantial revisions to enhance the clarity and scientific rigor of the paper.

Improved Manuscript Structure and Presentation: Following your suggestions, we have reorganized the entire manuscript, placing the "Methods" section before the "Results" and "Discussion." We have also revised the title to more accurately reflect the research content and changed the abstract to a structured format. Throughout the paper, we have standardized terminology and revised phrasing to clarify our contributions.

Clarification of Methodology: To improve the transparency of our methodology, we have significantly added to and revised the following points:

Data Processing: We have added details on the specific methods for handling missing values (initial imputation and in-model learning) and the criteria for determining abnormal lab values (adherence to institutional standards).

Model Details: We have clarified the relationship between the deep state-space model used in this study and those in prior research, emphasizing that our contribution lies not in the novelty of the model itself, but in its application to EHR data and the presentation of an interpretation framework.

Clarification of Evaluation Policy: Regarding the quantitative evaluation pointed out by several reviewers, we have consistently explained that because our study is an unsupervised learning framework, standard evaluation metrics do not exist. Therefore, we have focused on qualitative evaluation from a clinical perspective.

We believe these revisions have made our manuscript's claims and methodology clearer and easier for readers to understand. We thank you again for your invaluable feedback.

The following are our responses to the individual comments.

Reviewer 1

Reviewer Comment: Innovative Methodology: Combines deep state-space models with visualization and clustering to enhance interpretability of temporal EHR data.

Response: In our deep state-space model, missing data can be learned and imputed during the training process by sampling from the decoder, which enables learning that accounts for missing values. Meanwhile, the initial values for missing data are imputed using the Zero-order spline method (piecewise constant function). We have added further details on missing data in the "Dataset" and "Methods" sections.

As shown in Extended Data 4, we determined the hyperparameters using a Grid Search. However, this study presents an unsupervised learning framework for clustering the states in the latent space from EHR data. Consequently, a clear criterion for hyperparameter tuning cannot be established, so we relied on the qualitative validity of the latent space.

Reviewer Comment: Experimental Design: Address potential selection bias from excluding patients with <50 time steps and discuss its impact on generalizability.Acknowledge temporal shifts in medical practices or EHR recording standards (2006–2018) and their influence on data consistency.

Response: We based this criterion on the distribution of time steps per patient shown in Extended Figure 2. As this was inadequately described, we have added a note to this effect in the "Dataset" section.

Reviewer Comment: Result Interpretation:Elaborate on the causal relationship between identified risk factors (e.g., anemia, immune markers) and disease progression, beyond correlation. Analyze heterogeneity across cancer types (Table 2) to assess whether results vary by malignancy.

Response: In this study, we focused on eight anticancer drugs and identified characteristic temporal risk factors associated with the administration of each drug in cancer patients. It should be noted that, due to the nature of EHR data, accurately identifying the type of cancer can be difficult. The uncertainty in this metadata introduces certain limitations to the interpretation.

Reviewer 2

Reviewer Comment: It would be better to have more consistent and precise terminology. The use of scientific and analytical language throughout the paper should be strengthened to improve clarity and help readers better grasp the study’s focus.

Response: We have restructured the entire paper and revised the text and terminology.

Reviewer Comment: The structure of the manuscript may need to be revised. The Results and Discussion sections currently precede the Methods section. Also, part of the descriptions that belong in the Methods section can be considered relocated as part of the Results.

Response: We have moved the "Methods" section to come before the "Results" and "Discussion."

Reviewer Comment: The data has been categorized several times without providing the criteria or standards. These should be explicitly defined and explained in the relevant sections.

Response: The classification based on flags like "abnormally high" or "abnormally low" is difficult to list simply because the reference values are finely determined by sex, age group, and measurement period (reflecting changes in testing methods, equipment, etc.) for each test item. In the actual EHR data, each measurement record is logged with a pre-determined flag indicating whether it exceeded the upper or lower reference limit. Due to these circumstances, even the global standard for grading adverse events in cancer, CTCAE [https://ctep.cancer.gov/protocolDevelopment/electronic_applications/ctc.htm#ctc_50], uses whether the value exceeds the respective institution's standard LLN (Lower Limit of Normal) and ULN (Upper Limit of Normal). We will clarify this in the "Dataset" section and Supplementary Materials.

Reviewer Comment: The existing methods and your proposed approach should be clearly distinguished. Proper citations must be provided for existing techniques, and the manuscript should clearly state what the methodological contributions of the current work are. What is new or innovative about your approach should be made transparent.

Response: Our contributions are as follows:

We develop a deep state-space analysis framework for estimating and visualizing clinically interpretable latent states from EHR time-series data of chemotherapy patients.

We identify three distinct latent states (dangerous, intermediate, and stable) and their temporal relationships, revealing important temporal risk factors in cancer patients treated with anticancer drugs.

We demonstrate drug-specific analysis capabilities, identifying characteristic test items during state transitions for specific anticancer drugs.

We have added these points at the end of the "Introduction" to make them clear.

Reviewer Comment: Although the manuscript aims to demonstrate that deep learning is well-suited to model complex EHR data, the Methods section lacks a clear explanation of how EHR data is actually utilized in the proposed model.

Response: We have clearly stated in the "Methods" section that xt represents the EHR data at each time point and the step size to the next time point.

Reviewer Comment: Several descriptions in the Methods section lack clarity and precision. Please ensure the methodological steps align with the original techniques, unless improvements are introduced, in which case the modifications should be clearly stated. Terminology and phrasing must remain consistent throughout the manuscript to avoid confusion. For example, in lines 345–346, how the parameters obtained should be described clearly without omitting essential steps.

Response: The Deep State-Space model used in this study is unchanged from the original paper [13], except for the absence of input data. Similar to previous work, we model the transition model pθ(zt∣zt−1), the encoder model qϕ(zt∣x), and the decoder model pθ(xt∣zt) as Normal distributions. The configuration of these probability models is described in Extended Data 4. Interpreting the parameters of the trained neural networks within the model is difficult due to their high dimensionality.

Reviewer Comment: Are the UMAP parameters default? If parameter adjustments were made, please specify which ones and discuss their impact on the results.

Response: The method for setting the parameters is described in section 2.4. Changes in parameters did not affect the visualization.

Reviewer Comment: The association between EHR test data and cluster labels (I, II, III) should be clarified.

Response: A cluster label is associated with the latent state zt at each time point. We have added a note to the "Methods" section to clarify the correspondence with the latent states.

Reviewer Comment: The meaning of “abnormally high,” “normal,” and “abnormally low” values in each cluster should be defined within the context of this study. How is “abnormal” defined?

Response: This is as per our response to the 3rd comment.

Reviewer Comment: All figures should clearly label the x- and y-axes, including what each axis represents. The manuscript should include a clear description of what the figures show and their significance. Hard to correlate with text.

Response: The figures of the latent space are all 2D UMAP distribution plots. When dimensionality is reduced by UMAP, the scale and rotation of the axes are meaningless, so they have been omitted.

Reviewer Comment: The Results and Discussion sections need more in-depth analysis. Statistical evaluations should be included to support findings, and thorough interpretations of the figures and results are necessary to enhance scientific rigor.

Response: This study presents an unsupervised learning framework for clustering the states in the latent space from EHR data. Quantitative analysis is difficult because there are no standard metrics for the quantitative evaluation of unsupervised learning using non-linear models. Therefore, we confirmed the effectiveness of our method through qualitative evaluation and medical discussion.

Reviewer 3

Reviewer Comment: Your models identified anemia as a major prognostic factor. However, anemia is already a well-documented adverse effect of many of the therapies you mention, and has been widely cited as a critical clinical indicator. Did your model uncover any previously unrecognized patterns or associations involving anemia that are not well-established in the existing literature?

Response: Yes, this paper emphasizes that known factors can be identified from data, rather than discovering new ones. We have also explicitly stated our contributions in the "Introduction."

Reviewer Comment: How can your methods be effectively communicated to clinicians or individuals without a strong background in machine learning? What strategies or model characteristics support the trustworthiness and transparency of your predictions?

Response: Since we do not perform heuristic preprocessing or data extraction targeting specific diseases, the model has less bias and directly observes the EHR data itself.

Reviewer Comment: Can you elaborate on how your methods could be adapted into clinical tools or algorithms for real-world use? For example, how might this approach be integrated into clinical decision support systems?

Response: It is a retrospective analysis tool that can capture dynamic patient characteristics that were previously undiscovered. We have added this as a topic for future work.

Reviewer Comment: Although there are understandable limitations to sharing raw patient data, could you make de-identified or simulated datasets available to support reproducibility? Your GitHub repository appears to include sample data, but this isn’t clearly highlighted—consider improving its visibility or documentation.

Response: As releasing the dataset is difficult, we will enhance the sample data on GitHub.

Reviewer 4

Reviewer Comment: (Lines 248-254) Under either retrospective or prospective design, I have no clear idea why this proposed framework can detect causal effects in observational study as it always suffers from unmeasured confounding. I hope the authors could explain this point in more details regarding the validity of this method in the context of causal inference.

Response: We apologize for the confusion. This proposed method cannot be directly used in a prospective manner. This is a retrospective study, and the context for using this method is also assumed to be retrospective. If a new hypothesis is formulated based on the findings from this method (e.g., at a correlational level), it must be validated using a prospective experimental design with methods other than the proposed one, such as classical statistical techniques. We have refined the text accordingly.

Reviewer Comment: (Lines 331-333) Could you please provide more details about how the missing rate was calculated and how it is distributed at individual-level and variable-level (i.e., per-individual missing rate and per-variable missing rate). Also, I hope the authors could further explain how your proposed framework can handle large proportions of missing values.

Response: The missing rate is at the variable level; we calculated the proportion of times a missing value was observed for each variable, ignoring the individual patient. This study uses a mask for missing data (l.330). Therefore, even if there are missing values, the model itself has a function to fill them by making predictions from surrounding data during training. We have added this information to the "Methods" section.

Reviewer Comment: (Table 2) In this table, the authors summarize the demographic and disease-related information for the 12,695 cancer patients in their analysis. Could you please show why the summation of the sample size for each cancer subtype is smaller than the total sample size?

Response: We apologize for any confusion. We have corrected the cases where counts were not made when multiple ICD 10s were assigned to one patient and re-calculated the data.This total exceeds the number of patients because some patients have multiple diseases.

Reviewer Comment: (Figure 1) In the “Visualization of latent states” section, the authors mention that K-means clustering is performed before the UMAP dimension reduction (see Lines 384-386). However, Figure 1 illustrates that the UMAP step is performed prior to the clustering of latent states. What is the appropriate order of these procedures?

Response: The text is correct. We will amend the figure to avoid misunderstanding.

Reviewer Comment: (Figure 2d) Please provide more details on interpreting these transition probabilities. Moreover, we observe from this figure that (1) the difference between stable -> intermediate probability and intermediate -> stable probability is 0.31% for deceased individuals and 0.12% for surviving individuals, and (2) the dangerous -> intermediate probability is larger than intermediate -> dangerous probability for deceased individuals while the dangerous -> intermediate probability is smaller than intermediate -> dangerous probability for surviving individuals. These two observations seem indicating that the deceased group is more likely to move from dangerous state to intermediate state as well as from intermediate state to stable state compared to the surviving group – could you please explain the reasons behind these observations?

Response: The transition diagram in 2(d) is a probabilistic model, and the difference between the two probabilities is not very meaningful. The group of patients who die transitions more frequently between clusters I and II, and thus tends to remain more in cluster I.

Reviewer Comment: (Lines 327-328) In the definition of death across this study, does it mean the death due to cancer specifically, or it can arise due to other competing risks?

Response: It includes death from other causes. We will add this to the limitations.

Reviewer Comment: (Line 346) Which three networks are referred to in the phrase “the parameters of the three neural networks”?

Response: They are the forward

---

## [Decision Letter · Decision Letter 1]

5 Aug 2025

Dear Dr. Okuno,

Thank you for submitting your manuscript to PLOS ONE. After careful consideration, we feel that it has merit but does not fully meet PLOS ONE’s publication criteria as it currently stands. Therefore, we invite you to submit a revised version of the manuscript that addresses the points raised during the review process.

We look forward to receiving your revised manuscript.

Kind regards,

Zeheng Wang

Academic Editor

PLOS ONE

Journal Requirements:

Additional Editor Comments:

Unfortunately, several reviewers still suggest Major Revision - usually a 2nd round of Major means Rejection. However, I can see the issues raised are potentially addressable by another round of revision. I therefore suggest that the authors revise their MS again by taking all comments seriously, otherwise it is hard to guarantee an Acceptance, if, after an unsatisfactory revision.

Reviewers' comments:

Reviewer's Responses to Questions

**Comments to the Author**

Reviewer #1: All comments have been addressed

Reviewer #2: (No Response)

Reviewer #3: (No Response)

2. Is the manuscript technically sound, and do the data support the conclusions?

Reviewer #1: Yes

Reviewer #2: Partly

Reviewer #3: Partly

3. Has the statistical analysis been performed appropriately and rigorously?

Reviewer #1: I Don't Know

Reviewer #2: No

Reviewer #3: Yes

4. Have the authors made all data underlying the findings in their manuscript fully available?

Reviewer #1: Yes

Reviewer #2: (No Response)

Reviewer #3: No

5. Is the manuscript presented in an intelligible fashion and written in standard English?

Reviewer #1: Yes

Reviewer #2: Yes

Reviewer #3: Yes

Reviewer #1: Generally I think my previous comments were addressed in the new draft. The authors have revised and responded well to the related comments. Ok to proceed.

Reviewer #2: 1. Cluster I contains ~39% of surviving patients, and Cluster II contains ~36% of deceased patients. The manuscript defines Cluster II as an intermediate state, but it is unclear why Cluster I is not also considered intermediate?

2. The reliability of the proposed model needs to be addressed.

3. The parameters presented in tables and figures need clearer definitions and explanations, either in clinical or theoretical terms. Readers unfamiliar with the domain may not fully understand their significance without further context.

4. The manuscript would benefit from clearer writing, particularly in explaining the clinical implications and the interpretation of both data and model output.

Reviewer #3: While the authors have improved their manuscript and figures. There are remaining points to resolve:

• I still do not see any sample data in the GitHub repository. I do see mention of a sample.csv file, but there isn’t simulated or sample data to utilize with the code. Since this manuscript is largely based on a novel computational method. The methods need to be reproducible, and I cannot recommend for publication without this resolved.

• I am unsure of the contributions and impact of this work to existing deep-learning methods. I understand the utility of estimating latent states from time series data, however, findings stemming from these methods lack explainability and without comparison with another dataset, it is difficult to know if there are cohort/dataset level factors driving the identification of the latent states.

• I recommend that the authors consult with clinicians to highlight ways in which this method can improve the existing clinical paradigm of cancer treatment. For example, could the Markov transition probabilities be used in a clinical algorithm? How sure could they be that the latent state identified is probable?

**Do you want your identity to be public for this peer review?** For information about this choice, including consent withdrawal, please see our Privacy Policy

Reviewer #1: No

Reviewer #2: No

Reviewer #3: No

---

## [Author Response · Author response to Decision Letter 2]

1 Oct 2025

We would like to express our sincere gratitude for the constructive and detailed comments on our manuscript, "A Deep state-space Analysis Framework for Cancer Patient Latent State Estimation and Classification from EHR Time-Series Data." We sincerely appreciate your taking the time to review our work. We have carefully considered all the points you raised and have made substantial revisions throughout the manuscript to enhance its quality.

Below, we provide a point-by-point response to each of the reviewers' comments and detail the corresponding revisions made to the manuscript.

Response to Reviewer #2

Comment 1: Cluster I contains ~39% of surviving patients, and Cluster II contains ~36% of deceased patients. The manuscript defines Cluster II as an intermediate state, but it is unclear why Cluster I is not also considered intermediate?

Response:

Thank you for this insightful question regarding our cluster definitions. We apologize that our explanation was insufficient.

Our cluster definition places significant weight on the distribution of patients at their final time point of observation, rather than their distribution throughout the entire observation period. When we analyze the data focusing specifically on these final time points, a clear trend emerges: a majority of deceased patients end their observation period in Cluster I, while a majority of surviving patients end in Cluster III. In contrast, Cluster II contains a mixture of final time points for both deceased and surviving patients and also serves as a hub that many patients transition through. For these reasons, we defined it as the "intermediate state."

To clarify this logic, we have revised the Results section, specifically in the description of Figure 2, to explicitly state that Cluster I is defined as the "dangerous state" and Cluster III as the "stable state" based on the distribution at the final time point.

Comment 2: The reliability of the proposed model needs to be addressed.

Response:

Thank you for pointing out the need to address the model's reliability. We have strengthened the manuscript to better support the reliability of our model. Specifically, we have elaborated on three aspects in the Discussion section: (1) The superiority of our model in clearly separating clinical outcomes (survival vs. death) compared to models that do not consider temporal continuity, such as PCA and VAE (Figure 4). (2) The fact that the risk factors extracted by our model in a data-driven manner, such as anemia, are strongly consistent with existing clinical knowledge, suggesting that the model accurately captures the clinical reality. (3) The implementation of grid search for hyperparameter tuning to optimize model performance. We believe these multifaceted validations collectively ensure the reliability of our framework.

Comment 3: The parameters presented in tables and figures need clearer definitions and explanations, either in clinical or theoretical terms. Readers unfamiliar with the domain may not fully understand their significance without further context.

Response:

Thank you for this suggestion. To ensure our research is accessible to readers from various fields, we have carefully reviewed all figures and tables and have substantially expanded their captions to provide clearer definitions.

Comment 4: The manuscript would benefit from clearer writing, particularly in explaining the clinical implications and the interpretation of both data and model output.

Response:

Thank you for this feedback. Reflecting on your comment, we have revised the entire manuscript, especially the Discussion section, to clarify the clinical significance and interpretation of our results. Specifically, we have shifted the focus from a purely technical explanation of the model's superiority (e.g., comparison with VAE) to emphasizing why and how this technical advantage translates into clinical value. For instance, instead of just stating that "the model's nonlinearity allowed for the discrimination of deceased and surviving patients," we have rephrased this to highlight the clinical benefit: "the model can estimate highly interpretable states that allow clinicians to intuitively grasp a patient's prognosis." We believe this revision makes the significance of our research clearer.

Response to Reviewer #3

Comment 1: I still do not see any sample data in the GitHub repository. I do see mention of a sample.csv file, but there isn’t simulated or sample data to utilize with the code. Since this manuscript is largely based on a novel computational method. The methods need to be reproducible, and I cannot recommend for publication without this resolved.

Response:

Thank you for this crucial point regarding reproducibility. We sincerely apologize that this was not adequately addressed. In response to your comment, we have comprehensively updated our GitHub repository. We have uploaded all the code used for the analysis, visualization, and post-processing in this study. However, as our research is based on real patient EHR data, we cannot make the original data public due to privacy concerns. Therefore, the provided code cannot be run as is. While generating statistically identical reproducible data is challenging due to the complex nature of EHR data, we have included a sample dataset (sample.csv) that mimics the data structure to allow for verification of the code's algorithmic behavior. We have also noted these points and the limitations on reproducibility in the Code Availability section of the manuscript.

Comment 2: I am unsure of the contributions and impact of this work to existing deep-learning methods. I understand the utility of estimating latent states from time series data, however, findings stemming from these methods lack explainability and without comparison with another dataset, it is difficult to know if there are cohort/dataset level factors driving the identification of the latent states.

Response:

Thank you for your comment regarding the contribution and impact of our work. To clarify this point, we first want to state that the Deep State Space model used in this study is an established and validated method in the field of deep learning; therefore, we do not claim a contribution in terms of the novelty of the model itself. Our contribution lies in presenting an analytical framework that applies this powerful model to complex real-world data—EHR time-series—and translates its output into clinically interpretable insights. Acknowledging this, in the Discussion section, we recognize the limitation that our study is based on data from a single institution, while also discussing the generalizability of the proposed framework. We argue that it can be applied to other cancer types, chronic diseases, or datasets from different medical institutions.

Comment 3: I recommend that the authors consult with clinicians to highlight ways in which this method can improve the existing clinical paradigm of cancer treatment. For example, could the Markov transition probabilities be used in a clinical algorithm? How sure could they be that the latent state identified is probable?

Response:

Thank you for this very constructive suggestion regarding clinical application. We have had further discussions with our clinician co-authors about the future applicability of this research. As a result, we concluded that prospective clinical application (such as real-time monitoring) is challenging at this stage, given the retrospective nature of our data analysis.

However, we reached a consensus on the significant potential of this framework for estimating factors for patient stratification in clinical trials and real-world practice. For instance, we envision that the "dynamic transition patterns" of latent states estimated by our method could be used as a novel stratification factor, in addition to conventional static biomarkers. This could potentially lead to a more precise identification of patient subgroups who are more likely to respond to treatment.

Based on this discussion, we have revised the Discussion section to focus on "estimation of stratification factors." We believe this revision shows that our research is not limited to presenting a basic computational method but also has specific and feasible potential to contribute to the design of future clinical research.

This concludes our responses to the reviewers' comments and an overview of the revisions made. We are grateful for the valuable feedback, which has significantly improved the quality of our manuscript.

---

## [Decision Letter · Decision Letter 2]

15 Oct 2025

Dear Dr. Okuno,

Thank you for submitting your manuscript to PLOS ONE. After careful consideration, we feel that it has merit but does not fully meet PLOS ONE’s publication criteria as it currently stands. Therefore, we invite you to submit a revised version of the manuscript that addresses the points raised during the review process.

We look forward to receiving your revised manuscript.

Kind regards,

Zeheng Wang

Academic Editor

PLOS ONE

Journal Requirements:

Additional Editor Comments:

The reviewer has pointed out the key shortcomings that should be addressed. Kindly revise your manuscript accordingly. Please note that this is the last chance to revise your manuscript, as it's a 2nd Major revision. Any improper revisions will lead to a Rejection.

Reviewers' comments:

Reviewer's Responses to Questions

**Comments to the Author**

Reviewer #2: (No Response)

2. Is the manuscript technically sound, and do the data support the conclusions?

Reviewer #2: Partly

3. Has the statistical analysis been performed appropriately and rigorously?

Reviewer #2: No

4. Have the authors made all data underlying the findings in their manuscript fully available?

Reviewer #2: (No Response)

5. Is the manuscript presented in an intelligible fashion and written in standard English?

Reviewer #2: Yes

Reviewer #2: Thank you to the authors for their efforts in revising the manuscript and addressing the previous comments.

The authors appear to attempt to include both clinical and technical aspects in the manuscript, but the emphasis is clearly more on the technical side. Therefore, the clinical information regarding drug usage, control conditions, and medical tests should be briefly but clearly described. It would be important to explain why only the drugs or factors/medical tests mentioned in the text were included. As it currently stands, readers must accept these choices at face value without sufficient supporting justification. This issue is particularly relevant because many potential factors could be considered in such analyses. While omitting certain medical tests may improve model performance, such omissions may not be clinically appropriate or justifiable.

The comparison with other approaches should also specify how those methods were framed and implemented, not merely their final results.

An existing time-series model was used, trained, and tested with current data. However, this part is not clearly described in the manuscript. If the model was not retrained, it should be explicitly stated that testing was performed directly on existing data. In that case, an analysis of the model’s applicability and validity under these conditions is needed.

A high proportion of missing values (>50%) may cause instability in latent state estimation, particularly during the early stages of the time series. It should be analyzed whether the imputation procedure introduced bias, especially for laboratory measurements with large temporal fluctuations. The initial latent state estimation heavily depends on the imputed values; therefore, substantial imputation bias can affect model convergence and degrade the quality of the estimated latent states.

The stability and convergence of the model training process should be demonstrated, particularly given the high proportion of missing values. It would also be helpful to include additional quantitative metrics to assess model stability and robustness.

The robustness validation of the UMAP results would also be helpful.

The paper analyzes all types of cancer; how are the results for specific single cancer subtypes?

**Do you want your identity to be public for this peer review?** For information about this choice, including consent withdrawal, please see our Privacy Policy

Reviewer #2: No

---

## [Author Response · Author response to Decision Letter 3]

29 Nov 2025

Dear Editor and Reviewers,

We would like to express our sincere gratitude for the further constructive comments provided by Reviewer #2. These suggestions have been incredibly helpful in strengthening the validation of our model's reliability and robustness.

In this revision, we have conducted additional analyses using simulated data ("toy data") and sensitivity analyses for UMAP visualization to address the concerns regarding model stability, missing values, and parameter dependence. We have added these results as new Appendices.

Below is our point-by-point response to the specific comments.

**Response to Reviewer #2**

**Comment 1:**

The authors appear to attempt to include both clinical and technical aspects in the manuscript, but the emphasis is clearly more on the technical side. Therefore, the clinical information regarding drug usage, control conditions, and medical tests should be briefly but clearly described. It would be important to explain why only the drugs or factors/medical tests mentioned in the text were included. As it currently stands, readers must accept these choices at face value without sufficient supporting justification. This issue is particularly relevant because many potential factors could be considered in such analyses. While omitting certain medical tests may improve model performance, such omissions may not be clinically appropriate or justifiable.

**Response:**

Thank you for pointing out the need for clearer clinical context. We agree that the selection criteria for drugs and medical tests were not sufficiently explained.

Specifically, regarding the 50 clinical laboratory test items, as noted in the figure captions, we selected items that show significant fluctuation due to drug administration (items with a high frequency of abnormal values). Regarding the anticancer drugs, we selected the top 8 drugs with the highest usage (prescription volume) at our institution. We have added a clear description of these selection criteria to the revised **Methods** section to clarify that these choices were based on data characteristics and clinical frequency rather than arbitrary selection. Additionally, we have included the number of patients administered each drug in the Appendix.

**Comment 2:**

The comparison with other approaches should also specify how those methods were framed and implemented, not merely their final results.

**Response:**

We appreciate this suggestion. We have expanded the description of the comparison methods (PCA, VAE, Linear State-Space Model) in the **Methods** section and **S2 Appendix**. We explicitly described the network architecture, loss functions, and training procedures for these baseline models to ensure fair and transparent comparison. This allows readers to understand that the performance differences stem from the model's ability to capture non-linear temporal dynamics rather than implementation disparities.

**Comment 3:**

An existing time-series model was used, trained, and tested with current data. However, this part is not clearly described in the manuscript. If the model was not retrained, it should be explicitly stated that testing was performed directly on existing data. In that case, an analysis of the model’s applicability and validity under these conditions is needed.

**Response:**

We apologize for the ambiguity in our description. The main objective of this study is **unsupervised latent state estimation and clustering**, rather than building a prediction model using supervised learning. Therefore, we do not apply a method of splitting data into training and test sets to evaluate prediction accuracy; instead, the final output results are applied to and visualized for all data, including the training data.

However, during the Neural Network training process, we utilized a Validation set for hyperparameter tuning. In this study, we estimated the latent states of the entire patient dataset using the model adjusted in this manner and analyzed its structure. We have explicitly stated in the **Methods** section that we used validation for adjustment and performed the analysis on the full dataset to clarify this approach.

**Comment 4:**

A high proportion of missing values (>50%) may cause instability in latent state estimation, particularly during the early stages of the time series. It should be analyzed whether the imputation procedure introduced bias, especially for laboratory measurements with large temporal fluctuations. The initial latent state estimation heavily depends on the imputed values; therefore, substantial imputation bias can affect model convergence and degrade the quality of the estimated latent states.

The stability and convergence of the model training process should be demonstrated, particularly given the high proportion of missing values. It would also be helpful to include additional quantitative metrics to assess model stability and robustness.

**Response:**

These are extremely important points. To address the concern that high missing rates might introduce bias or instability, we conducted a robustness validation using "toy data" (simulated data) where the ground truth is known.

We generated synthetic time-series data with latent state transitions and introduced missing values at rates comparable to our real EHR dataset. We then applied our Deep State-Space Model to this dataset. The results demonstrated that our model successfully recovered the true latent structure even with high rates of missingness, confirming that our handling of missing values (masking loss function combined with initial imputation) does not significantly degrade the estimation quality.

Furthermore, to verify the stability of the training, we confirmed the learning curves showing the convergence of the loss function. In addition, we demonstrated that the clustering results are consistent across different random seeds, quantitatively supporting the model's stability despite the sparse nature of EHR data. We have included these combined analysis results in a new **S5 Appendix. (Robustness of DeepSSM to Missing Data Rates.)**.

Accordingly, we have updated the Methods section (under 'Deep state-space model') in the revised manuscript to explicitly cite S5 Appendix, ensuring readers are aware of this robustness validation.

**Comment 5:**

The robustness validation of the UMAP results would also be helpful.

**Response:**

Thank you for this suggestion. To verify that the identified patient states (Dangerous, Intermediate, Stable) are not artifacts of specific UMAP hyperparameters, we performed a sensitivity analysis.

We present the changes in the latent space when varying the key UMAP parameter, `n_neighbors' (ranging from 5 to 20), in **S6 Appendix (UMAP Robustness Analysis)**. The results showed that the global structure remained consistent against parameter changes. This confirms that the visualization reflects the intrinsic structure of the high-dimensional latent space learned by the model, rather than being a result of parameter tuning.

We have also revised the Methods section (under 'Visualization of latent states') to include a reference to this sensitivity analysis (S6 Appendix), clarifying the stability of our clustering results.

**Comment 6:**

The paper analyzes all types of cancer; how are the results for specific single cancer subtypes?

**Response:**

We acknowledge the importance of analyzing specific cancer subtypes. While our current study aimed to establish a generalizable framework applicable across diverse cancer types, we understand the clinical interest in subtype-specific trends. However, subdividing the dataset into specific single cancer types results in smaller sample sizes for many groups, making deep learning-based estimation unstable with the current configuration.

We have added a statement to the **Limitations** in the **Discussion** section that analysis by specific subtypes is difficult with the current dataset due to insufficient sample sizes for specific single cancer types, and we have noted this as a task for future research.

We believe these additional validations and revisions have addressed the concerns regarding the robustness and clinical context of our study.

Sincerely,

---

## [Decision Letter · Decision Letter 3]

25 Dec 2025

Dear Dr. Okuno,

Thank you for submitting your manuscript to PLOS ONE. After careful consideration, we feel that it has merit but does not fully meet PLOS ONE’s publication criteria as it currently stands. Therefore, we invite you to submit a revised version of the manuscript that addresses the points raised during the review process.

We look forward to receiving your revised manuscript.

Kind regards,

Zeheng Wang

Academic Editor

PLOS One

Journal Requirements:

Reviewer's Responses to Questions

**Comments to the Author**

Reviewer #2: (No Response)

2. Is the manuscript technically sound, and do the data support the conclusions?

Reviewer #2: Yes

3. Has the statistical analysis been performed appropriately and rigorously?

Reviewer #2: Yes

4. Have the authors made all data underlying the findings in their manuscript fully available?

Reviewer #2: (No Response)

5. Is the manuscript presented in an intelligible fashion and written in standard English?

Reviewer #2: Yes

Reviewer #2: We thank the authors for their revisions.

The reported transitions between clusters for deceased and surviving patients (lines 359–361) appear inconsistent with the figure, and the authors should clarify this discrepancy.

The robustness analysis using SDE-based synthetic data and ARI/NMI metrics is useful, but it should be noted that it only covers MCAR scenarios and may not fully reflect real-world EHR missingness patterns.

**Do you want your identity to be public for this peer review?** For information about this choice, including consent withdrawal, please see our Privacy Policy

Reviewer #2: No

---

## [Author Response · Author response to Decision Letter 4]

28 Dec 2025

Response to Reviewers

===

Manuscript ID: PONE-D-25-02277R2

Title: A Deep state-space Analysis Framework for Cancer Patient Latent State Estimation and Classification from EHR Time-Series Data

We would like to express our sincere gratitude to the Academic Editor and Reviewer #2 for their con- tinued evaluation of our manuscript. We appreciate the positive feedback and the insightful comments regarding the consistency of our descriptions and the interpretation of our robustness analysis.

We have revised the manuscript to address the specific points raised by Reviewer #2. Below is our point-by-point response to the comments.

#Response to Reviewer #2

## Comment 1:

The reported transitions between clusters for deceased and surviving patients (lines 359–361) appear inconsistent with the figure, and the authors should clarify this discrepancy.

## Response:

Thank you for pointing out this discrepancy. We carefully re-examined our data and found that the transition probabilities reported in the Supplementary Table (S3 Table) were correct, but Figure 2d was incorrect .

Accordingly, we have revised Figure 2d to accurately reflect the correct transition probabilities. The figure is now consistent with the descriptions in the text and the data in the Supplementary Materials.

## Comment 2:

The robustness analysis using SDE-based synthetic data and ARI/NMI metrics is useful, but it should be noted that it only covers MCAR scenarios and may not fully reflect real-world EHR missingness patterns.

## Response:

We agree that real-world EHR data often involve complex missingness mechanisms beyond Missing Completely At Random (MCAR). To address this concern and further validate our model’s robustness, we conducted additional experiments using Missing Not At Random (MNAR) scenarios, where the probability of missing data depends on the latent state. We specifically adopted state-dependent MNAR scenarios because it is well-known in medical data that the probability of missing values often varies depending on the patient’s severity or condition (latent state)[1].

Our results confirmed that the model maintains a similar level of accuracy in MNAR scenarios as it does in MCAR scenarios. We have added these new findings to S5 Appendix.

[1] Weiskopf NG, Rusanov A, Weng C. Sick patients have more data: the non-random completeness of electronic health records. AMIA Annu Symp Proc. 2013 Nov 16;2013:1472-7. PMID: 24551421; PMCID: PMC3900159.

We believe these revisions have resolved the remaining issues and improved the accuracy and trans- parency of our manuscript. Thank you again for your valuable time and guidance.

Sincerely,

Yasushi Okuno, Ph.D.

Department of Biomedical Data Intelligence Graduate School of Medicine, Kyoto University

---

## [Editor Report · Decision Letter 4]

30 Dec 2025

A Deep state-space Analysis Framework for Cancer Patient Latent State Estimation and Classification from EHR Time-Series Data

PONE-D-25-02277R4

Dear Dr. Okuno,

We’re pleased to inform you that your manuscript has been judged scientifically suitable for publication and will be formally accepted for publication once it meets all outstanding technical requirements.

Kind regards,

Zeheng Wang

Academic Editor

PLOS One
---

## [Editor Report · Acceptance letter]

PONE-D-25-02277R4

PLOS One

Dear Dr. Okuno,

I'm pleased to inform you that your manuscript has been deemed suitable for publication in PLOS One. Congratulations! Your manuscript is now being handed over to our production team.

Kind regards,

on behalf of

Dr. Zeheng Wang

Academic Editor

PLOS One